# Intercomparison of upper tropospheric and lower stratospheric water vapor measurements over the Asian Summer Monsoon during the StratoClim Campaign

Clare E. Singer[1,a], Benjamin W. Clouser[1], Sergey M. Khaykin[2], Martina Krämer[3], Francesco Cairo[4], Thomas Peter[5], Alexey Lykov[6], Christian Rolf[3], Nicole Spelten[3], Armin Afchine[3], Simone Brunamonti[5,b], and Elisabeth J. Moyer[1]

[1]Dept. of the Geophysical Sciences, University of Chicago, Chicago, IL, USA
[2]LATMOS, UVSQ, Sorbonne Université, CNRS, IPSL, Guyancourt, France
[3]Forschungszentrum Jülich, Institut für Energie und Klimaforschung (IEK-7), Germany
[4]National Research Council of Italy, Institute of Atmospheric Sciences and Climate (CNR-ISAC), Rome, Italy
[5]Institute for Atmospheric and Climate Science, ETH Zürich, Zürich, Switzerland
[6]Central Aerological Observatory of RosHydroMet, Dolgoprudny, Russian Federation
[a]now at: Dept. of Environmental Science and Engineering, California Institute of Technology, Pasadena, CA, USA
[b]now at: Empa, Laboratory for Air Pollution/Environmental Technology, Dübendorf, Switzerland

**Correspondence:** C. E. Singer (csinger@caltech.edu) and E. J. Moyer (moyer@uchicago.edu)

**Abstract.** In situ measurements in the climatically important upper troposphere / lower stratosphere (UTLS) are critical for understanding controls on cloud formation, the entry of water into the stratosphere, and hydration/dehydration of the tropical tropopause layer. Accurate in situ measurement of water vapor in the UTLS however is difficult because of low water vapor concentrations ($< 5$ ppmv) and a challenging low temperature/pressure environment. The StratoClim campaign out of Kathmandu, Nepal in July and August 2017, which made the first high-altitude aircraft measurements in the Asian Summer Monsoon (ASM), also provided an opportunity to intercompare three in situ hygrometers mounted on the M-55 Geophysica: ChiWIS (Chicago Water Isotope Spectrometer), FISH (Fast In situ Stratospheric Hygrometer), and FLASH (Fluorescent Lyman-$\alpha$ Stratospheric Hygrometer). Instrument agreement was very good, suggesting no intrinsic technique-dependent biases: ChiWIS measures by mid-infrared laser absorption spectroscopy and FISH and FLASH by Lyman-$\alpha$ induced fluorescence. In clear-sky UTLS conditions ($H_2O < 10$ ppmv), mean and standard deviations of differences in paired observations between ChiWIS and FLASH were only $(-1.4 \pm 5.9)\%$ and those between FISH and FLASH only $(-1.5 \pm 8.0)\%$. Agreement between ChiWIS and FLASH for in-cloud conditions is even tighter, at $(+0.7 \pm 7.6)\%$. Estimated realized instrumental precision in UTLS conditions was 0.05, 0.2, and 0.1 ppmv for ChiWIS, FLASH, and FISH, respectively. This level of accuracy and precision allows the confident detection of fine-scale spatial structures in UTLS water vapor required for understanding the role of convection and the ASM in the stratospheric water vapor budget.

# 1 Introduction

Water vapor is one of the most important gases in Earth's atmosphere because of its control on dynamics and interactions with radiation. Water in Earth's atmosphere interacts with longwave radiation in both the vapor and condensed phases. In the vapor phase, $H_2O$ is a greenhouse gas that roughly doubles the anthropogenic warming from carbon dioxide alone (Dessler et al., 2008). Ice crystals in high-altitude cirrus clouds both trap outgoing longwave radiation as well as scatter incoming shortwave radiation. In the atmosphere, water vapor also controls large-scale atmospheric flows and convection through latent heating. The net radiative effects of clouds are a balance between shortwave reflection (cooling from low and high clouds) and longwave absorption (heating from high clouds). Furthermore, changes in cloud distributions or amounts can change large-scale atmospheric circulation, like the Hadley Cell, by perturbing the atmospheric heating profile (Schneider et al., 2010).

Amounts of water vapor in the upper troposphere and lower stratosphere (UTLS) are quite small, usually below 5 parts per million by volume (ppmv). Because the mixing ratio of $H_2O$ in the UTLS is so low, small absolute changes have very large relative effects. Water vapor in the UTLS has several important effects including the direct radiative effect as a greenhouse gas (warming) (Solomon et al., 2010), indirect radiative effect through formation of cirrus clouds (cooling of surface and warming of upper levels of the atmosphere) (Lee et al., 2009), and also influences stratospheric ozone chemistry (Vogel et al., 2011). Furthermore, the stratospheric water vapor feedback (i.e., the increase of stratospheric water vapor with global mean temperature) is one of the largest positive climate feedbacks that acts to amplify warming (Dessler et al., 2013).

The Asian Summer Monsoon (ASM) is known to be one of the largest regional sources of $H_2O$ to the stratosphere (Dethof et al., 1999; Kremser et al., 2009; Randel et al., 2012). The ASM also transports short-lived chemicals including $NO_x$ and VOCs and aerosol particles from the surface to the UTLS through its active convection (Randel and Park, 2006; Randel et al., 2010). These pollutants, and $H_2O$, have relatively long lifetimes in the stratosphere and are known to deplete stratospheric ozone. The StratoClim measurement campaign in July and August 2017 made the first in situ aircraft measurements of these trace gases and particles in the ASM anticyclone UTLS.

Measurements in the tropical UTLS are very challenging because it is such a remote region and difficult to access. At $15 - 20$ km altitude, it is only accessible for in situ measurements with balloons and a select number of specialized aircraft, including the M-55 Geophysica, the platform for the StratoClim aircraft campaign. Measurements of water vapor in the UTLS are further complicated because the mixing ratios are so low. When concentrations are only 5 ppmv, absolute precision of 1 ppmv still translates to uncertainties of 20%. Furthermore, for understanding cloud processes, it is necessary to have even greater accuracy of vapor measurements because small changes in supersaturation have significant impacts on clouds (Jensen et al., 2005; Jensen and Pfister, 2005; Jensen et al., 2008; Krämer et al., 2009).

Although it has long been recognized that measuring water vapor at high altitudes is challenging (Oltmans et al., 2000), rigorous intercomparison studies of in situ $H_2O$ measurements, like this, are still critical for creating clear and interpretable scientific results. Discrepancies between in situ measurements (on aircraft and balloon) and satellite measurements have been documented and studied for decades (Oltmans et al., 2000; Vömel et al., 2007; Weinstock et al., 2009; Rollins et al., 2014; Meyer et al., 2015; Hall et al., 2016; Kaufmann et al., 2018). Even very small disagreements in the absolute humidity in the

UTLS (1 ppmv compared to a background of 3 ppmv) can corresponds to differences in measured relative humidity of $> 30\%$. This has significant implications for understanding ice microphysical processes (Jensen et al., 2005; Jensen and Pfister, 2005; Peter et al., 2006; Jensen et al., 2008). A previous in situ comparison in Vömel et al. (2007) found that measurements between the balloon-borne cryogenic frostpoint hygrometer (CFH), Harvard Lyman-$\alpha$ hygrometer (HWV), FLASH (Lyman-$\alpha$), and NOAA frost point hygrometer were as large as 10–20% (altitude dependent) even though the combined instrument uncertainties of these instruments was only 5–10%. Weinstock et al. (2009) compared HWV with CFH and the NOAA frost point hygrometer and found there was a systematic bias of $1 - 1.5$ ppmv and differences up to 30% in the UTLS. More recently, Rollins et al. (2014) compared $H_2O$ measurements taken during the NASA MACPEX campaign over Houston, Texas by HWV, JLH (TDL), ALIAS (TDL), FISH (Lyman-$\alpha$), DLH (TDL), and (CIMS)-$H_2O$ (mass spectrometry). They found differences in mixing ratios of up to 20% (0.8 ppmv). They cited how these discrepancies in $H_2O$ measurements complicated the interpretation of in-cloud $RH_{ice}> 130\%$ and clear-sky $RH_{ice}> 160\%$ (above homogeneous nucleation threshold). Meyer et al. (2015) reviewed measurements from numerous field campaigns with the FISH instrument and found that over two decades the agreement between measurements from FISH and other instruments in the $< 10$ ppmv range improved from $\pm30\%$ to $\pm5 - 20\%$. Vömel et al. (2016) and Hall et al. (2016) both conducted studies of balloon-borne hygrometers, finding that technological advances have improved the agreement between instruments on simultaneous launches. Most recently, Kaufmann et al. (2018) did an intercomparison study of $H_2O$ measurements made during the ML-CIRRUS campaign on the DLR HALO aircraft in 2014 over central Europe. They compared AIMS (mass spectrometry), FISH, and SHARC (TDL) $H_2O$ measurements in the UTLS ($< 20$ ppmv) and found that they agreed within their combined uncertainty of $\pm10 - 15\%$, depending on humidity range. The mean values during the campaign agreed within 2.5%, although systematic differences of $10-15\%$ were found during the driest periods below 10 ppmv. Instrument intercomparison studies have also been conducted in controlled cloud chamber settings to mitigate the technical challenges of high-altitude flight measurements. Of note are the AquaVIT experiments performed at the AIDA cloud chamber. Fahey et al. (2014) describes the results from AquaVIT-1 in 2007 (AquaVIT-2 and -3 are not yet published). The core instruments compared were APicT (TDL), CFH, FISH-1 and FISH-2, FLASH-B1 and FLASH-B2 (Lyman-$\alpha$), HWV, and JLH. These instruments agreed within $\pm20\%$ for the $1 - 150$ ppmv range. Importantly however, the conditions within a controlled cloud chamber like this cannot replicate flight conditions and these experiments cannot replace in situ intercomparison studies such as the one here.

In this study we present an intercomparison between the three in situ hygrometers on-board the Geophysica aircraft during the StratoClim campaign. The campaign was conducted during the summer of 2017 over the ASM region from Kathmandu, Nepal. The hygrometers include the new ChiWIS integrated cavity output spectrometer and the established Lyman-$\alpha$ vapor and total water hygrometers, FLASH and FISH. We first compare paired water vapor measurements taken by the three hygrometers during the flights, and briefly discuss two case studies. We further analyze the relative humidity measurements as one way to constrain the absolute accuracy of the hygrometers. Lastly, we conclude by comparing the in situ aircraft measurements with measurements made during a simultaneous balloon campaign out of nearby Dhulikhel, Nepal (Brunamonti et al., 2018) and satellite measurements from the Microwave Limb Sounder instrument. We use the in situ measurements to validate the satellite retrievals, and discuss the high-resolution details observed in the aircraft data but lacking in the satellite observations.

 ## 2 StratoClim Campaign Overview and Instrument Descriptions

The StratoClim aircraft field campaign was conducted during July and August 2017 out of Kathmandu, Nepal. The goals of the campaign were to sample the upper levels of the ASM anticyclone and quantify the amount of transport of near-surface air and pollutants to the UTLS. This coordinated aircraft and balloon campaign made the first in situ measurements from aircraft of the ASM, though previous balloon measurements have been made in the area (e.g., Bian et al., 2012; Vernier et al., 2018; Ma et al., 2022). The aircraft campaign was comprised of eight flights between July 27 and August 10, 2017 using the M-55 Geophysica research platform. The flights will be referred to as F$x$, with $x$ being the flight number, in the remainder of the paper. The flight paths and altitude profiles are shown in Fig. 1a and b, respectively. Fig. 1c shows water vapor profiles from six flights during the campaign between 350 and 480 K potential temperature and $2 - 200$ ppmv.

The Geophysica payload during StratoClim included three instruments measuring water vapor which allows for an opportunity to compare different instrument measurement methods. Two independent methods were used: integrated cavity output absorption spectroscopy (ChiWIS) and Lyman-$\alpha$ photofragment fluorescence spectroscopy (FLASH and FISH). A summary of the three hygrometers is given in Table 1. Time series and vertical profiles of $H_2O$ and relative humidity as measured by the three hygrometers throughout the flight campaign are shown in Figs. S1–S4.

**Table 1.** Summary of realized instrument performance* measured in UTLS conditions for the three in situ hygrometers

| Instrument | Technique | Measured quantity | Resolution [Hz] | Range [ppmv] | Precision [ppmv] |
|---|---|---|---|---|---|
| ChiWIS | TDL OA-ICOS | $H_2O$ | 1 | 1–100 | 0.05 |
| FLASH | Lyman-$\alpha$ | $H_2O$ | 1 | 1–1000 | 0.2 |
| FISH | Lyman-$\alpha$ | Total $H_2O$ | 1 | 1–1000 | 0.1 |

*Precision values as measured during an 8-minute, constant altitude segment of F4 (cf. Fig. S5).

### 2.1 ChiWIS

The Chicago Water Isotope Spectrometer (ChiWIS) is a new flight instrument designed for airborne measurements of vapor-phase water isotopologues in the dry UTLS. A previous version designed for chamber measurements, ChiWIS-lab, is described in Sarkozy et al. (2020). The flight version of the ChiWIS is a tunable diode laser (TDL), off-axis integrated cavity output spectrometer (OA-ICOS) designed to measure primarily HDO and $H_2O$ at stratospheric mixing ratios. The spectrometer scans absorption lines of both species near 2.647 $\mu$m wavelength in a single current sweep. The flight instrument is mounted on top of the Geophysica aircraft and uses a rear-facing inlet to measure only vapor phase water. Its 90 cm cell length and $R = 0.9998$ reflectivity mirrors provide an effective path length of better than 7 km, with little deviation in reflectivity during the campaign. The effective pathlength is measured for each flight using periodic in-flight measurements of cavity ringdown time. Table 1 gives 1-s precision of $H_2O$ of 0.05 ppmv. For isotopic ratio measurements at 10-s integration, realized precision

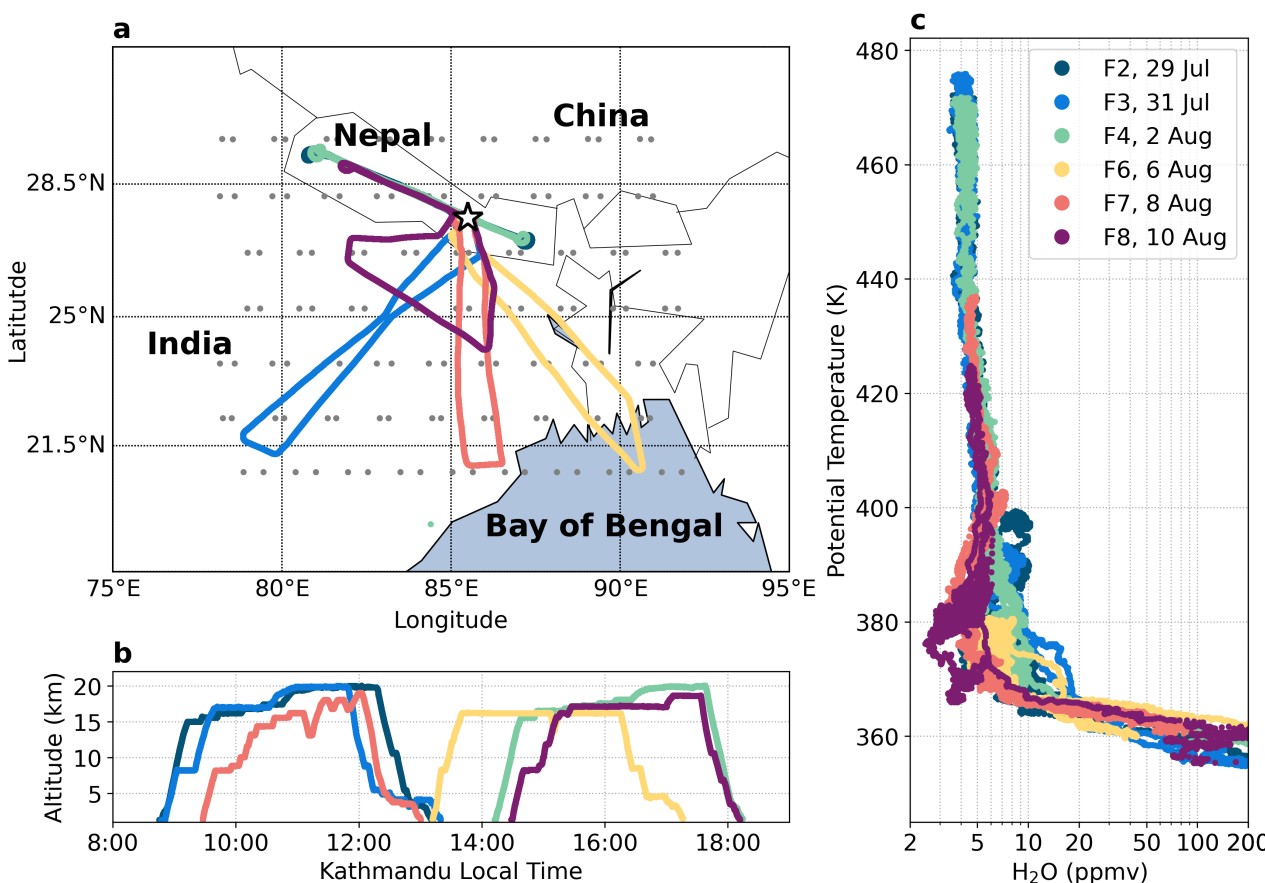

**Figure 1.** a) Map of of the ASM region with flight tracks overlaid in solid colored lines. Balloonsonde launch point (Dhulikhel, Nepal) is shown with the star symbol and locations of MLS v5 profiles are shown in grey dots. The aim of flight F6 was to measure the convective outflow from a typhoon that had occurred in the days prior and the outflow was reaching the very edge of the aircraft's range over the Bay of Bengal on August 6. F8 was specifically sampling a very local strong convective storm over India. b) Altitude flight profiles shown in local Kathmandu time. Flights F2, F3, and F7 took place during the morning, while F4, F6, and F8 occurred during the afternoon during which there is generally more active convection. F2–F4 included long legs at high altitude above the tropical tropopause layer (TTL) for the remote sensing instruments. F7 was targeting the detailed structure of the TTL and included several V-shaped profiles in the later half of the flight. c) Profiles of $H_2O$ measured by FLASH against potential temperature for each of the six flights (F2–F4, F6–F8).

is 18 ppbv and 80 pptv for $H_2O$ and HDO, respectively. We report here on 1-Hz data. Before the flight, the instrument is flushed with dry air and the inlet is kept sealed until the aircraft reaches $\approx 300$ hPa to avoid contamination of the measurement cell with moist tropospheric air. Post-processing of the raw spectra includes a laser "pedestal" correction (to remove stray light) procedure before spectral line fitting with unmodified Hitran parameters (Clouser et al., 2022, in prep.). We exclude here the highest-altitude periods of the flights (roughly 70 mbar or below in ambient pressure) where the internal cell pressure

110

of ChiWIS, regulated at 40 mbar, lost regulation and dropped below 30 mbar. In these conditions desorption of water vapor from the optical cavity walls produces a noticeable effect in measurements. During the StratoClim campaign, ChiWIS reported measurements for six of the eight scientific flights (all except F1 and F5).

## 2.2 FLASH

FLASH-A (Fluorescent Lyman-$\alpha$ Stratospheric Hygrometer for Aircraft) (Sitnikov et al., 2007), designed specifically for the M55-Geophysica aircraft, is the airborne version of the FLASH-B balloon-borne hygrometer. The instrument was redesigned in 2009 (Khaykin et al., 2013) and significantly improved and updated again for the StratoClim flights. Unlike the previous versions of FLASH-A with transversal optical setup, the version flown during StratoClim employs a coaxial optics similar to the balloon-borne version of FLASH (Yushkov et al., 1998). FLASH-A is mounted under the right wing of the aircraft and has a rear-facing inlet designed to measure only the vapor phase. The chamber is maintained at a constant temperature (24°C) and pressure (36 hPa) and the inlet tube is heated to 30°C. Before the flight, the instrument is ventilated for several hours using dry air ($< 1$ ppmv) and the inlet is kept sealed until the aircraft reaches 250 hPa to avoid chamber contamination with moist tropospheric air. The turnover time of air in the measurement chamber is 0.19 s and during the StratoClim flights FLASH reported measurements averaged to a 1 Hz sampling frequency. The precision on the 1 Hz data in the stratosphere is 0.2 ppmv with a detection limit of 0.1 ppmv for a 5 s integration time. FLASH-A was calibrated against a reference MBW-373LX frost-point hygrometer before and after the aircraft deployment as well as during the campaign using FISH calibration facility (Zöger et al., 1999). During the StratoClim campaign, FLASH reported measurements for all eight scientific flights as well as during the transfer flight to Kathmandu.

## 2.3 FISH

The Fast In situ Stratospheric Hygrometer (FISH) is also a Lyman-$\alpha$ fluorescence spectrometer. FISH has a forward-facing inlet and measures total water (gas-phase plus evaporated ice particles) at a rate of 1 Hz in the range 1–1000 ppmv (Zöger et al., 1999; Meyer et al., 2015). The outer and inner inlet tubes are heated to 90 and 70 °C, respectively, to ensure a complete evaporation of the sampled ice crystals. FISH is calibrated regularly in the laboratory and in the field between flights to the reference frost-point hygrometer MBW-373LX or DP30, which is integrated in an automated calibration bench (Meyer et al., 2015). The flow through the measuring cell is enabled only at ambient pressure below 350–400 hPa in order to prevent moisture from entering the tubing at lower altitudes. To ensure a high precision measurement, the intensity of the Lyman-$\alpha$ lamp and also the lamp background counts are recorded every 12 seconds. For mixing ratios down to 1 ppmv, the uncertainty reaches a lower limit of 0.3 ppmv. Because FISH measures total water, a direct comparison with ChiWIS and FLASH is only possible during periods of clear-sky. In a recent review of the FISH instrument, it was noted that discrepancies between FISH and FLASH during clear-sky were sometimes greater than 100%, but usually less than 30% (Meyer et al., 2015); during the StratoClim flights, these discrepancies were much smaller. During the StratoClim campaign, FISH reported measurements for five of the eight scientific flights (all but F1, F3, and F5).

## 2.4 Temperature and pressure

Full meteorological data (pressure, temperature, altitude) were measured on-board the Geophysica by the aircraft aeronautical system (UCSE) (Stefanutti et al., 1999) and temperature and pressure by a separate scientific instrument, the Rosemount thermodynamic complex (TDC) (Shur et al., 2006). In the field, systematic differences between the temperatures measured by UCSE and TDC at high altitudes appeared to be driven by discrepancies in the calculated Mach number. During post-processing the TDC temperature was recalculated using the Mach number from UCSE. Temperature and pressure measurements are used in this analysis to calculate the saturation vapor pressure, saturation specific humidity, and relative humidity with respect to ice ($RH_{ice}$) according to Murphy and Koop (2005). We use TDC temperatures with Mach correction because of their high temporal resolution (1 Hz). Estimated accuracy and precision are 0.5 K and 0.1 K, respectively, and dominate uncertainty in relative humidity. The measurement uncertainty on temperature alone (assuming a temperature of 200 K and a perfect measurement of $H_2O$ and pressure) translates to a fractional uncertainty ($\Delta RH_{ice}$ / $RH_{ice}$) of about 0.08. Conversely, a measurement uncertainty from $H_2O$ alone would need to be as large as 0.4 ppmv at a background stratospheric value of 5 ppmv to produce the same fractional uncertainty in derived relative humidity.

## 2.5 Balloon CFH

In conjunction with the Geophysica flights, StratoClim organized a simultaneous balloon campaign in Nepal, which is discussed in detail in Brunamonti et al. (2018). 11 balloon launches with the Cryogenic Frostpoint Hygrometer (CFH) on-board were made during the period August 3–12, 2017 from Dhulikhel, Nepal, roughly 20 km East of Kathmandu airport. We construct a mean balloonsonde profile from these 11 launches for comparison in this paper. The CFH (Vömel et al., 2007, 2016) uses the "chilled-mirror" technique to measure the ambient water vapor concentrations with an uncertainty of 10% up to 28 km altitude. A temperature-controlled mirror is exposed to the air while an optical detector senses the presence of condensate on the mirror. The mirror temperature is adjusted until the point where the mirror maintains constant reflectivity and the amount of condensate can be assumed to be in equilibrium with the gas-phase. This temperature, the dew/frost point temperature, is measured with a thermistor and the specific humidity is calculated. Occasional artifacts can be produced in CFH measurements after encounters with mixed-phase clouds if supercooled droplets freeze in the inlet tube of the instrument and subsequently re-evaporate in the dry stratosphere (Jorge et al., 2020). Potentially contaminated data were rejected from the analysis as described in Brunamonti et al. (2019).

## 2.6 MLS

The Microwave Limb Sounder (MLS) instrument, operating onboard the NASA Aura satellite, measures vertical profiles of temperature and several trace gas species. Here we use 126 water vapor profiles spatially and temporally co-located with the StratoClim flights as a point of comparison (shown in Fig. 1a). We use version 5.0 (v5) profiles which were selected in the region between 20–30°N and 78–92°E during the campaign dates of 27 July – 10 August 2017, using screening criteria from Livesey et al. (2022). We also show MLS version 4.3 (v4) profiles (only 118) which were selected using screening criteria in

Livesey et al. (2020). We interpolate the $H_2O$ profiles onto a potential temperature grid using the MLS temperature product provided at the same pressure levels. MLS v5 includes a correction on the $H_2O$ retrievals described in Livesey et al. (2021), which results in an approximately spatially uniform drying at 68 hPa of about 15% compared to v4.

## 2.7 Excluded data

The intercomparison is primarily based on measurements made from the three in situ hygrometers on-board the Geophysica aircraft during the StratoClim campaign. Comparisons are restricted to F2–F4 and F6–F8 because only the FLASH instrument reported data for F1 and F5. Table 2 summarizes the total hours of data included in the intercomparison, removing exclusion described in this section. Hours of measurements collected from each instrument are reported for in-cloud, clear-sky, and total.

**Table 2.** Summary of total hours measured on each flight in UTLS conditions ($< 10$ ppmv). Hours included in the intercomparison (as described in Section 2.7) are reported in total, in-cloud, and in clear-sky for each instrument.

| Flight | Total hours (ChiWIS / FLASH / FISH) | Hours in-cloud (ChiWIS / FLASH / FISH) | Hours in clear-sky (ChiWIS / FLASH / FISH) |
|---|---|---|---|
| F2 | 1.59 / 2.98 / 2.32 | 0.00 / 0.00 / – | 1.59 / 2.98 / 2.32 |
| F3 | 1.07 / 2.23 / – | 0.08 / 0.08 / – | 0.99 / 2.15 / – |
| F4 | 1.77 / 2.90 / 0.86 | 0.71 / 0.73 / – | 1.07 / 2.17 / 0.86 |
| F6 | 2.52 / 2.48 / 0.45 | 1.62 / 1.61 / – | 0.89 / 0.87 / 0.45 |
| F7 | 1.22 / 1.23 / 0.28 | 0.83 / 0.79 / – | 0.39 / 0.44 / 0.28 |
| F8 | 1.72 / 2.13 / 0.11 | 1.35 / 1.35 / – | 0.37 / 0.78 / 0.11 |
| Total | 9.89 / 13.95 / 4.01 | 4.60 / 4.56 / – | 5.29 / 9.39 / 4.01 |

This paper focuses on comparisons in the UTLS so we restrict our comparison to measurements between 12 and 20 km. We also remove ascent and descent periods where the aircraft was moving vertically at a rate faster than 10 m s$^{-1}$ because these fast changes in altitude exacerbate small differences in the timing of measurements and the location of the instruments on the aircraft due to sharp vertical gradients in $H_2O$.

ChiWIS was designed to operate with the cell pressure at approximately 40 mbar to maintain a 0.5 s flush time. When the ambient pressure dropped too low, the pump was unable to maintain pressure inside the cell and the flush time increased, allowing for possible adsorption of $H_2O$ onto the cavity walls and subsequent desorption, which may have artificially increased the reported values. Due to this, the main intercomparison excludes ChiWIS measurements made when the cell pressure was below 30 mbar. For completeness, periods where the cell pressure was between 20 and 30 mbar are shown in Figs. S1–S4, and several key figures in the subsequent analysis are duplicated in the supplement including all periods where cell pressure was above 20 mbar.

Data from FISH taken only during periods of clear-sky are included in the intercomparison. Since FISH measures total water (vapor and condensed phases), while ChiWIS and FLASH measure only water vapor, a one-to-one comparison can only be done between the three hygrometers during flight periods of clear-sky. The definition of clear-sky is explained below.

Finally, four periods from F8 have been excluded from the analysis. During F8, the plane flew through four very active overshooting convective towers, and both ChiWIS and FLASH inhaled ice particles despite their rear-facing inlets. Because these time periods do not represent vapor-only measurements, they were not reported by either instrument. Clipping of these periods of particle inhalation was done independently by the two groups based on a combination of anomalously wet and exponentially decaying $H_2O$ signals, MAS backscatter ratio data, and NIXE-CAPS ice particle number concentration.

We also note that the three hygrometers have different physical sampling rates, in addition to reported measurement frequency. All measurements are first interpolated to a common grid and then compared one-to-one. FISH has a sampling rate of 1 s. ChiWIS is limited to a maximum resolution of approximately 0.5 s due to the flush time of the large optical cavity, but data is averaged to 1 s to increase the signal-to-noise ratio; for HDO longer averaging intervals (2–10 s) are used. FLASH on the other hand has a very small chamber with a flush time of a fraction of 1 s and the time resolution is limited by the averaging interval necessary to increase the signal to noise ratio sufficiently for the desired measurement precision.

### 2.7.1 Definition of clear-sky

Clear-sky periods are defined by an absence of ice particles as measured by two independent instruments. We use the backscatter ratio (BR) from the Multiwavelength Aerosol Scatterometer (MAS) (Cairo et al., 2011) and the ice particle number concentration ($N_{ice}$) from the Novel Ice EXpEriment - Cloud and Aerosol Particle Spectrometer (NIXE-CAPS) (Meyer, 2012; Krämer et al., 2016, 2020b). Clear-sky periods are defined as when $N_{ice} = 0\ cm^{-3}$ and $BR < 1.2$ with a lag time that is flight-dependent. Fig. 2 shows time series measurements of $N_{ice}$ from each flight with a binary cloud flag determined from BR, $N_{ice}$, and the ice water content (IWC) overlaid. IWC is a product derived as the difference between FISH and FLASH measurements (or derived from the particle size distribution when FISH or FLASH are unavailable) (Krämer et al., 2020b), and thus is an unreliable flag for clear-sky in an intercomparison study like this where we are specifically interested in the small deviations in vapor measurements between these two instruments. In general, these three metrics agree well, and we define clear-sky as periods when $N_{ice}$ and BR are both below the threshold.

A flight-dependent time lag ($\tau_c$) is applied to the cloud flag to account for timing discrepancies between instruments and saturation of the FISH measurement chamber. When the FISH instrument is exposed to very high IWCs it takes a finite amount of time (order 10 s) for the chamber to clear out and report accurate vapor measurements again. The optimal $\tau_c$ was chosen to remove outlying measurements, and can be visualized in Fig. S6. When $\tau_c = 0$, the structure of the differences between FISH and FLASH are skewed right, meaning positive differences of FISH measuring wetter than FLASH are more common, which is due to erroneous in-cloud measurements being included in the sample. The lag $\tau_c$ is chosen for each flight such that the differences are roughly symmetric, and we can be confident that the differences reported are truly comparing vapor-only measurements.

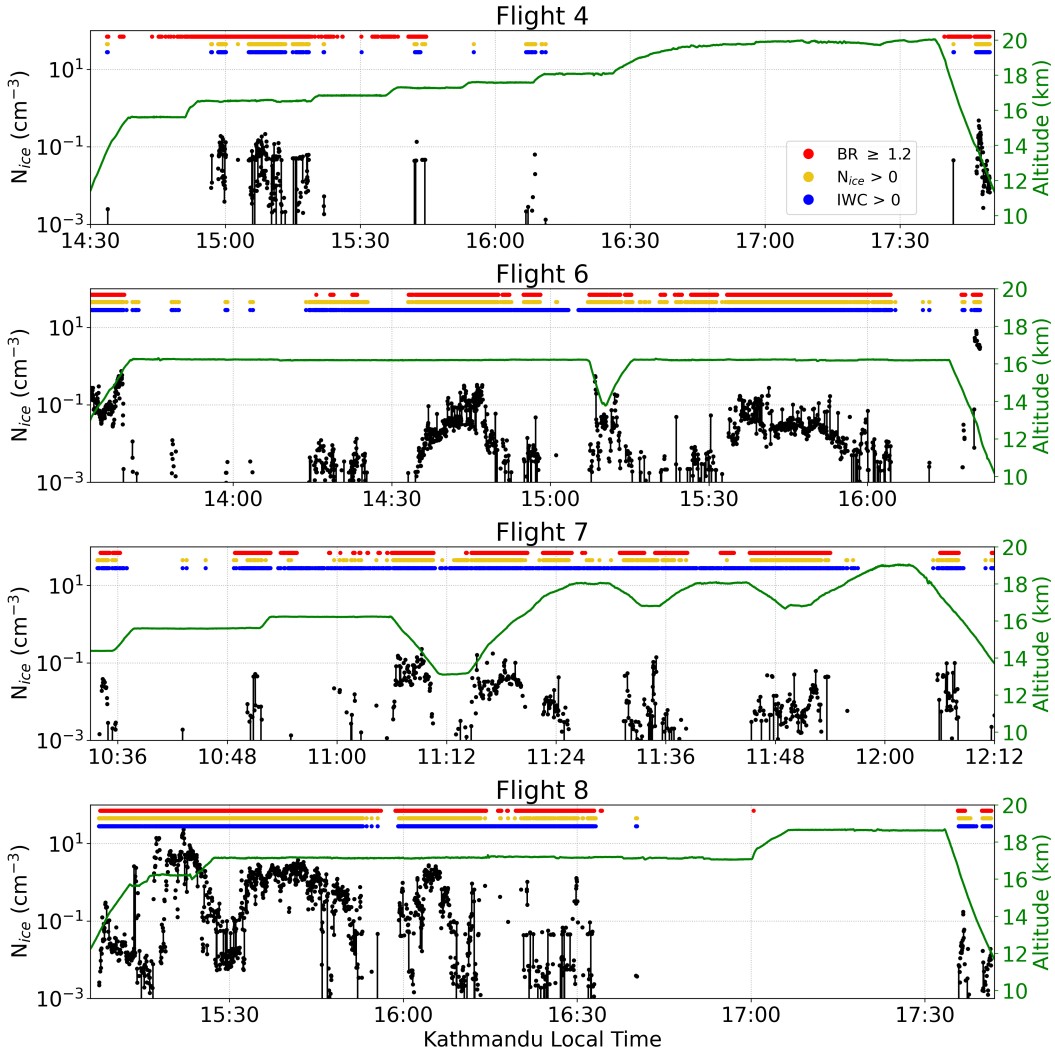

**Figure 2.** Time series of ice particle number concentration ($N_{ice}$, black) measured by NIXE-CAPS for F4, F6, F7, and F8. No clouds were sampled on F2 or F3. Shown in colors at the top of each panel are cloud flags derived from MAS backscatter ratio (BR $\geq$ 1.2, red), $N_{ice}$ > 0 cm$^{-3}$ (yellow), and ice water content produce (IWC > 0 g kg$^{-1}$, blue). Clear-sky is inversely defined as regions where BR < 1.2 or $N_{ice}$ = 0 cm$^{-3}$. The IWC is defined as the difference between FISH and FLASH, and is thus an unreliable flag for clear-sky in an intercomparison study like this where we are specifically interested in the small deviations in vapor measurements between these two instruments. Flight altitude is shown in green.

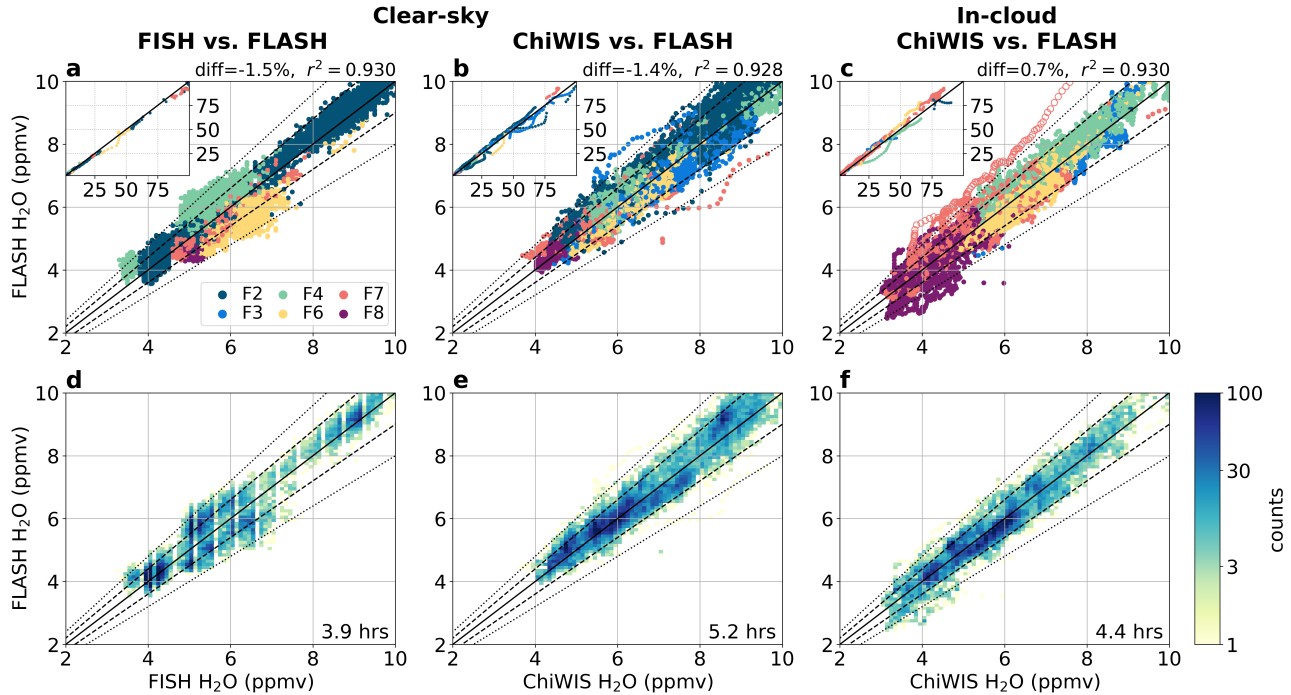

**Figure 3.** $H_2O$ vapor correlations between the three in situ hygrometers in the stratospheric range of 2–10 ppmv: a) clear-sky FISH vs. clear-sky FLASH, b) clear-sky ChiWIS vs. clear-sky FLASH, c) in-cloud ChiWIS vs. in-cloud FLASH. Insets show a larger range from 2–100 ppmv. Points are colored by flight number and plotted in random order. The open circles in panel c) on F7 mark the time period of disagreement between ChiWIS and FLASH as the airplane was ascending out of a deep dive. Panels d)–f) show the same information as a)–c) but as the frequency of observations over all the flights in each 0.1 ppmv by 0.1 ppmv bin. The total hours of paired observations (for $H_2O < 10$ ppmv) is shown in the bottom right corner. The one-to-one line is plotted in solid black with $\pm 10$ and $\pm 20\%$ shown in dashed and dotted lines. The percentage difference and $r^2$ coefficients are shown above each panel.

## 3 In situ water vapor measurements

### 3.1 Overview

As an overview we show a point-by-point comparison between the three in situ hygrometers for the entire campaign, color-coded by flight. Figure 3 shows scatter plots of ChiWIS vs. FLASH and clear-sky FISH vs. FLASH from 2–10 ppmv (with the inset showing 2–100 ppmv).

The figure includes all measurements from the three hygrometers other than those excluded as detailed in section 2.7. One additional period is excluded from statistical analysis of instrument differences: the ascent after the dive on flight F7, marked on Fig. 3 by open circles. During this ascent, FLASH reported a substantially wetter measurement than ChiWIS, likely associated with drying out after the deep, wet dive. In-cloud periods where ChiWIS cell pressure is poorly regulated $(20 - 30$ mbar) are shown in Fig. S7.

240 Instrument agreement is generally excellent. Mean percentage difference between instruments at UTLS levels ($< 10$ ppmv) is calculated as $\Delta = \overline{\left(\frac{x-y}{y}\right)} \times 100\%$ and shown in Table 3 for all instrument pairs. Across the whole campaign, instrument agreement between pairs is better than 2% in all cases. Differences for clear-sky FISH and FLASH, clear-sky ChiWIS and FLASH, and in-cloud ChiWIS and FLASH are $(-1.5 \pm 8.0)\%$, $(-1.4 \pm 5.9)\%$, and $(+0.7 \pm 7.6)\%$, respectively. Correlations between these same instrument pairs are $r^2 = 0.930$, $0.928$, and $0.930$, respectively, and the fraction of individual measure-

245 ments agreeing to better than $\pm 10\%$ are 78%, 92%, and 87%. In wetter conditions (2–100 ppmv), analogous mean differences are $(-1.3 \pm 8.0)\%$, $(-1.4 \pm 6.2)\%$, and $(+0.3 \pm 7.7)\%$ and correlations are $r^2 = 0.993$, $0.987$, and $0.994$.

**Table 3.** Summary of measurement differences between instrument pairs for UTLS conditions ($< 10$ ppmv). Mean percentage differences ($\Delta$) and standard deviations ($\sigma_\Delta$) between simultaneous measurements for each pair is given for each flight individually, the two periods of the campaign, and the campaign in total. ChiWIS and FLASH comparisons are further broken in clear-sky, cloudy, and all periods.

|  | F2 | F3 | F4 | F6 | F7 | F8 | warm/wet | cold/dry | all |
|---|---|---|---|---|---|---|---|---|---|
| FISH vs. FLASH (clear-sky) | -2.3 (4.4) | – | -10.1 (4.3) | 11.5 (5.9) | 6.0 (4.3) | 10.8 (4.0) | -4.4 (5.6) | 9.6 (5.8) | -1.5 (8.0) |
| FISH vs. ChiWIS (clear-sky) | 4.4 (4.9) | – | -6.4 (4.0) | 8.2 (4.3) | 8.4 (10.2) | 3.5 (2.9) | -0.8 (7.0) | 8.2 (6.9) | 1.9 (8.1) |
| ChiWIS vs. FLASH (clear-sky) | -4.9 (4.9) | 2.4 (5.3) | -3.0 (4.4) | 3.1 (3.6) | -1.9 (7.7) | -2.6 (4.1) | -2.3 (5.7) | 0.7 (5.7) | -1.4 (5.9) |
| ChiWIS vs. FLASH (cloudy) | -3.8 (1.8) | 3.0 (6.7) | -3.9 (4.2) | 3.1 (3.9) | -2.1 (7.4) | 1.9 (10.6) | -3.3 (4.9) | 1.6 (7.8) | 0.7 (7.6) |
| ChiWIS vs. FLASH (all) | -4.9 (4.9) | 2.4 (5.4) | -3.4 (4.3) | 3.1 (3.8) | -2.1 (7.5) | 1.0 (9.8) | -2.5 (5.6) | 1.3 (7.2) | -0.4 (6.8) |

*Values in table shown as $\Delta(\sigma_\Delta)$, the mean and standard deviations of percentage differences: $\Delta = \text{mean}\left[\left(\frac{x-y}{y}\right) \times 100\%\right]$ and $\sigma_\Delta = \text{std}\left[\left(\frac{x-y}{y}\right) \times 100\%\right]$.

 Some flight-to-flight variations are seen, most evidently in FISH (Fig. 3a). Points for F2 and F4 generally fall above the 1:1 line (FISH drier than FLASH), while points from F6–F8 fall below the 1:1 line (FISH wetter than FLASH). These flight-to-flight variations can be more easily seen in Table 3 or Fig. 4, which shows histograms of the difference between measured

250 $H_2O$ between either ChiWIS and FLASH or clear-sky FISH and FLASH for each flight. When examining the performance of FISH compared to FLASH, flight-to-flight variations tend to cancel out when averaged across the campaign: in Table 3 (first row), the mean difference during the first "warm/wet" half of the campaign compared to the second "cold/dry" half were $-4.4\%$ and $+9.6\%$, which averaged together meant only a -1.5% mean difference across the whole campaign, but with a sizable variance ($\pm 8.0\%$). For ChiWIS and FLASH the flight-to-flight variations in instrument performance were much smaller, which

255 is evident from the smaller standard deviation in percentage differences across the campaign, even though the mean difference of the clear-sky comparison was very similar to FISH.

 The median difference between each instrument pair (during clear-sky periods with $H_2O < 10$ ppmv) is shown in Fig. 4 and ranges from -0.6 ppmv to +0.48 ppmv for FISH and -0.41 ppmv to +0.23 ppmv for ChiWIS. These differences are likely

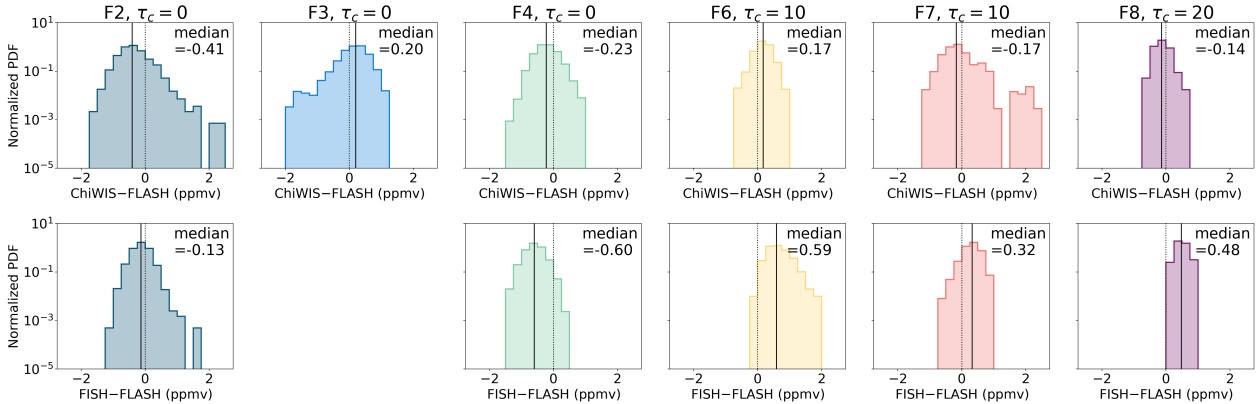

**Figure 4.** Normalized PDF of absolute differences in paired clear-sky observations below 10 ppmv for FLASH and ChiWIS (top row) or FISH (bottom row). The 10-ppmv cutoff is set by FLASH. FISH did not report measurements for F3. The time lag $\tau_c$ applied to the cloud flag for each flight is indicated in the title. The dotted line shows zero mean difference and the solid line shows the median difference for each flight; this value is marked in the top right corner of each subplot. On F2 and F4, FISH median difference from FLASH is negative, while on F6–F8 it is positive; this trend is also distinguishable in Fig. 3. The spread in measurement differences between paired observations on each flight is consistent with the joint precision of the instruments ($\approx$ 0.2–0.4 ppmv).

significant – with clear-sky measurements accounting for 20-95 minutes per flight (20-100%), see Table 2 – though they could be related to differences in flight profiles. The smallest number of clear-sky datapoints occurs in the heavily convective F8, which also shows the smallest instrumental median differences.

In addition to analyzing each flight separately, we also analyze the differences between the three instruments aggregated over all six flights. Fig. 5 shows the vertical profile (against potential temperature) of the percentage difference between each two sets of measurements shown in Fig. 3. The number of observations in each 2% by 2 K bin is shown by the color, with the color scale plotted logarithmically to highlight small discrepancies between the instruments. The mean differences (and standard deviations) from FLASH between the instruments are $(-1.3 \pm 8.0)\%$, $(-1.4 \pm 6.2)\%$, and $(+0.3 \pm 7.7)\%$ for clear-sky FISH, clear-sky ChiWIS, and in-cloud ChiWIS, respectively (also shown in Fig. 5). The differences between clear-sky FISH and FLASH (Fig. 5a) exhibits some vertical structure, while the differences between ChiWIS and FLASH (clear-sky and in-cloud, Figs. 5b and c, respectively) are quite vertically uniform.

## 3.2 Individual flights

As done in Khaykin et al. (2022), we naturally break the campaign into two periods, the first, observed during flights F2–F4, was "warm/wet", while the second, observed during flights F6–F8, was "cold/dry." The coldest and driest periods of the campaign were associated with more clouds (see the difference between Fig. 3b and c). Fig. 6 shows flights F2 and F7 as examples from the warm/wet and cold/dry periods, respectively. On both flights we see excellent agreement between the three hygrometers; all are able to capture the fine-scale variability. F2 (warm/wet) sampled two wet layers ($\sim$ 10 ppmv) above the

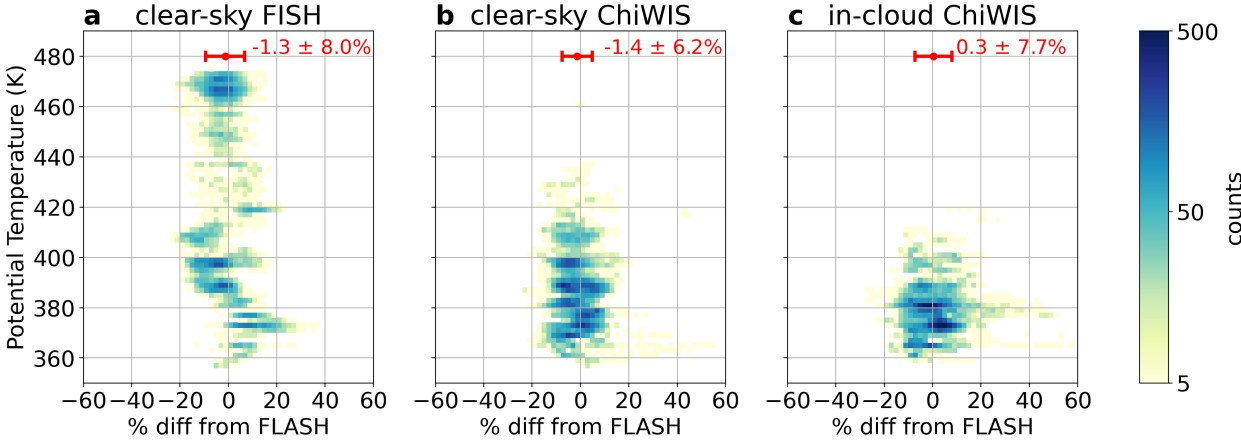

**Figure 5.** Histogram of percentage difference in paired $H_2O$ observations between instruments as a function of potential temperature for a) clear-sky FISH, b) clear-sky ChiWIS, c) in-cloud ChiWIS, all compared to FLASH. The colors show the number of observations in each 2% by 2 K bin. The red error bars indicate the mean and one standard deviation of the percentage difference between the two instruments. Apparent altitude-dependent structure in FISH vs. FLASH may relate to flight-to-flight differences. Positive deviations in ChiWIS in cloudy conditions around 380 K are related to extreme ice concentrations in fresh anvil outflow on F8.

cold-point tropopause (CPT) around 390 and 399 K. F7 (cold/dry) sampled around the CPT in detail and measured several cirrus clouds in situ. In contrast to F2, the lowest $H_2O$ mixing ratios observed on F7 were down to around 3 ppmv, again with all three hygrometers agreeing very well on the magnitude and spatial variability of $H_2O$. See Khaykin et al. (2022) for a more detailed study of F2 and F7. Time series and profiles of $H_2O$ for all flights can be found in Figs. S1 and S2.

F2 took place in the morning on July 29 over Nepal. The flight pattern consisted of stair steps up from 15 km to nearly 18 km at approximately 500 m intervals with a long flight leg at 20 km (altitude shown in green, Fig. 6b). Note that during the highest altitude leg on F2 the ChiWIS cell pressure dropped below 30 mbar (the measurements are shown by grey dots for completeness but excluded from the main analysis). During the stair stepping, there were two moist layers around 17 and 18 km or 390 and 400 K (10:10–10:25 and 10:25–10:45 am local time, respectively). These layers had similar $H_2O$ mixing

ratios, around 10 ppmv, but occurred at different altitudes and were shown to have different origin (Khaykin et al., 2022). Also notable is the high spatial/temporal variability of the water vapor in the second moist layer. This variability is clearly physical, and our confidence in this is due to the remarkable agreement seen by the three hygrometers over this stretch. Also of note is the very high precision of the ChiWIS instrument compared to both FISH and FLASH, which can be seen by the very small amplitude high frequency variations that we attribute to measurement noise.

F7 took place in the morning on August 8 and the flight path went due south from Kathmandu over India. This flight pattern was designed to robustly sample the UTLS and the tropopause, so the aircraft did a seesaw pattern between 17 and 19 km during the second half of the flight (after a deep dive). The water vapor around the CPT was observed to be highly variable on

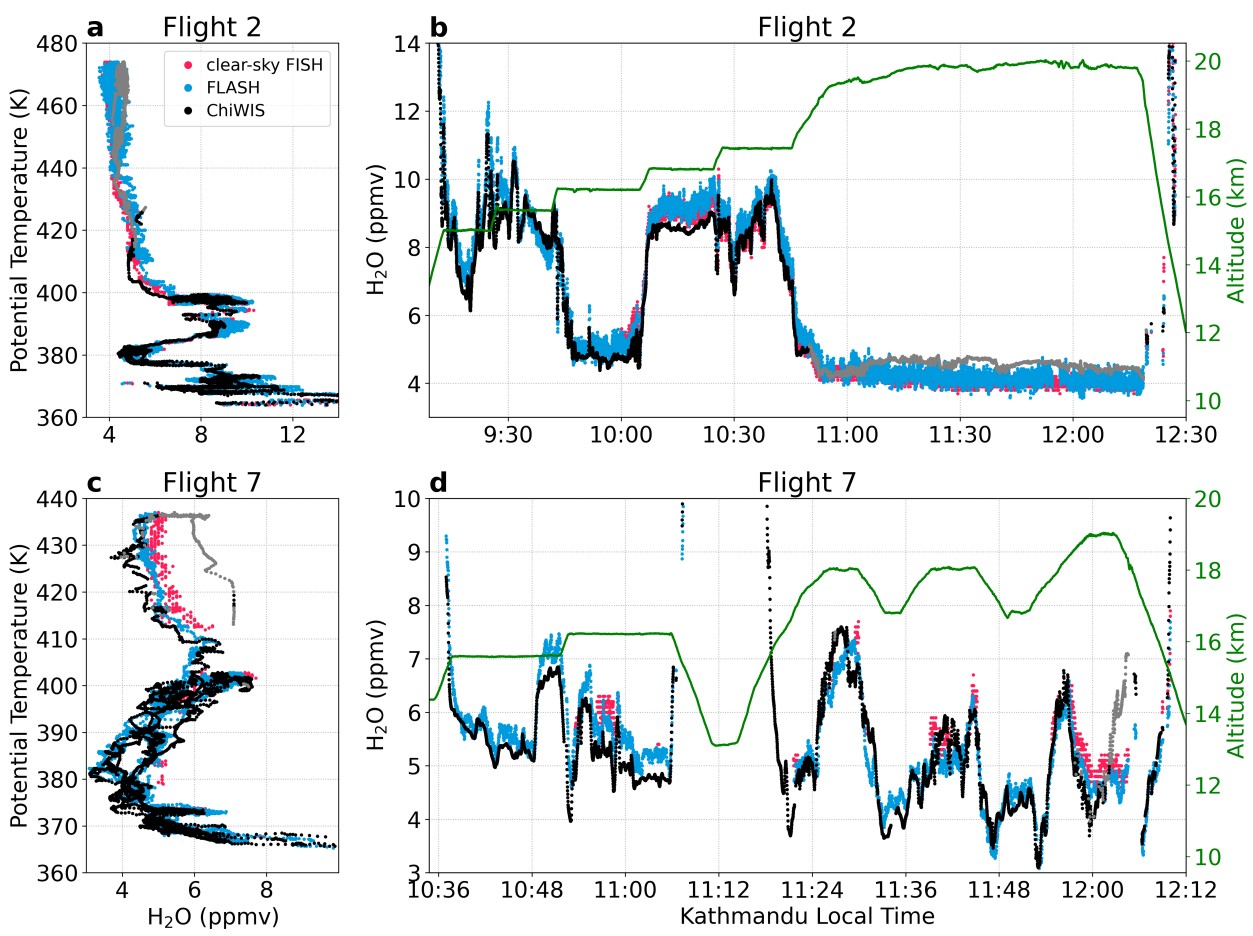

**Figure 6.** Vertical profiles (a, c) and time series (b, d) of water vapor from F2 (a-b) and F7 (c-d). Due to differing flight strategies on F2 and F7, vertical range is smaller for F7 (c-d) show more detail. Altitude (green) and in situ $H_2O$ measurements from ChiWIS (black/grey), FLASH (blue), and clear-sky FISH (pink). Periods where ChiWIS cell pressure is just out of regulation, between 20 and 30 mbar, are excluded from the intercomparison but shown in grey for reference. Note, this was common during the later parts of F2–F4 and F8 when the aircraft flew long level flight legs at nearly 20 km. Across both of these example flights, there is impressive agreement between the three hygrometers, to the point that it is difficult to even pick out the FISH measurements plotted beneath the other two. The higher precision of ChiWIS creates sharper apparent temporal structure in the time series (b, d). Nevertheless, all three instruments are able to capture fine-scale variability in atmospheric water vapor. F2 (a-b) shows an example of two very wet layers ($\sim$ 10 ppmv) above the cold-point tropopause (CPT) around 390 and 399 K with agreement between the three hygrometers about the magnitude and spatial variability of these layers. F7 (c-d) shows an example of variability of $H_2O$ around the CPT with mixing ratios ranging from 3–7.5 ppmv related to passage through cirrus clouds.

a scale of a few hundred kilometers (Khaykin et al., 2022). The data gaps in clear-sky FISH measurements on F7 in Fig. 6d are due to the presence of in situ cirrus clouds.

 **4 Relative humidity in clear-sky and clouds**

We use the relative humidity derived from $H_2O$ and temperature/pressure measurements as an important, independent metric to assess the performance of the in situ hygrometers. Unlike in liquid clouds, relative humidity with respect to ice ($RH_{ice}$), can deviate significantly from 1 due to thermodynamic inertia and kinetic limitations of the growth of ice crystals. Although supersaturations are expected, they are bounded by the homogeneous nucleation threshold, above which we do not expect

to find measurements, since ice crystals will form. Because the temperature and pressure measurements are also subject to uncertainties and errors, this is not an absolute measure of the in situ hygrometer accuracy. However, by comparing ChiWIS, FLASH, and FISH, we can make physically informed statements about the performance about the three hygrometers.

Fig. 7 shows the frequency of measurements made for a given value of $RH_{ice}$ and temperature for all three hygrometers both in clear-sky and in-cloud. The measurements are put in bins of 0.5 K for temperature by 0.05 for $RH_{ice}$ and the frequencies are

305 standardized. It is common to construct a two-dimensional histogram of relative humidity as a function of in situ temperature (e.g., Krämer et al. (2020b) figure 10 which also show clear-sky and in-cloud $RH_{ice}$ distributions from FLASH), because in temperature vs. $RH_{ice}$ space, there is a theoretical limit due to homogeneous nucleation (dashed line, Koop et al. (2000)). Fig. S8 shows the same but includes periods when ChiWIS cell pressure is between 20 and 30 mbar.

In clear sky periods, the mean relative humidity measured by the three hygrometers is 0.79, 0.51, and 0.52, for ChiWIS,

FLASH, and FISH, respectively. ChiWIS reports an anomalously high mean value for clear-sky $RH_{ice}$ because the driest measurements were made at very high altitude where the cell pressure was unregulated and those data have been removed. If these data are included then the ChiWIS mean relative humidity in clear-sky is 0.51 (see Fig. S8). For in-cloud periods, only ChiWIS and FLASH report measurements of relative humidity, with mean values of 1.07 and 1.07, respectively. Including the unregulated periods does not alter the in-cloud mean relative humidity because there were very few clouds sampled at

sufficiently high altitudes where ChiWIS cell pressure was unregulated. All measurements indicate that the most frequently sampled part of this phase space (ignoring the very dry high-altitude flight legs) was around 200 K at just below $RH_{ice} = 1$ for clear-sky and just above $RH_{ice} = 1$ for cloudy-sky.

Both ChiWIS and FLASH report very infrequent, but non-zero, measured points above the homogeneous nucleation threshold at very low temperatures ($T < 190$ K) during in-cloud periods. This may be attributed to measurement uncertainty on either

the $H_2O$ or temperature values, both of which are very difficult to measure at these low mixing ratios and cold temperatures. Another explanation may be that in the nucleation phase of an ice cloud, the small crystals have not grown large enough due kinetic limitations on vapor uptake, which prevents the $RH_{ice}$ from reducing further; this phenomenon has been previously called "peak $RH_{ice}$" (Kärcher and Lohmann, 2002; Krämer et al., 2009).

Fig. 8 shows a point-by-point comparison of $RH_{ice}$ measurements made by the three hygrometers. Similar features as in

Fig. 3 are also noticeable here, like the secular trend in differences between clear-sky FISH and FLASH going from negative to positive over the course of the six flights. Other features are more apparent when viewed in $RH_{ice}$ space, rather than $H_2O$ space though, like the large discrepancies of in-cloud measurements on F8 between ChiWIS and FLASH or the subsaturated streamer on F4 (Fig. 8c). The mean percentage difference and $r^2$ correlation values of clear-sky FISH, clear-sky ChiWIS, and

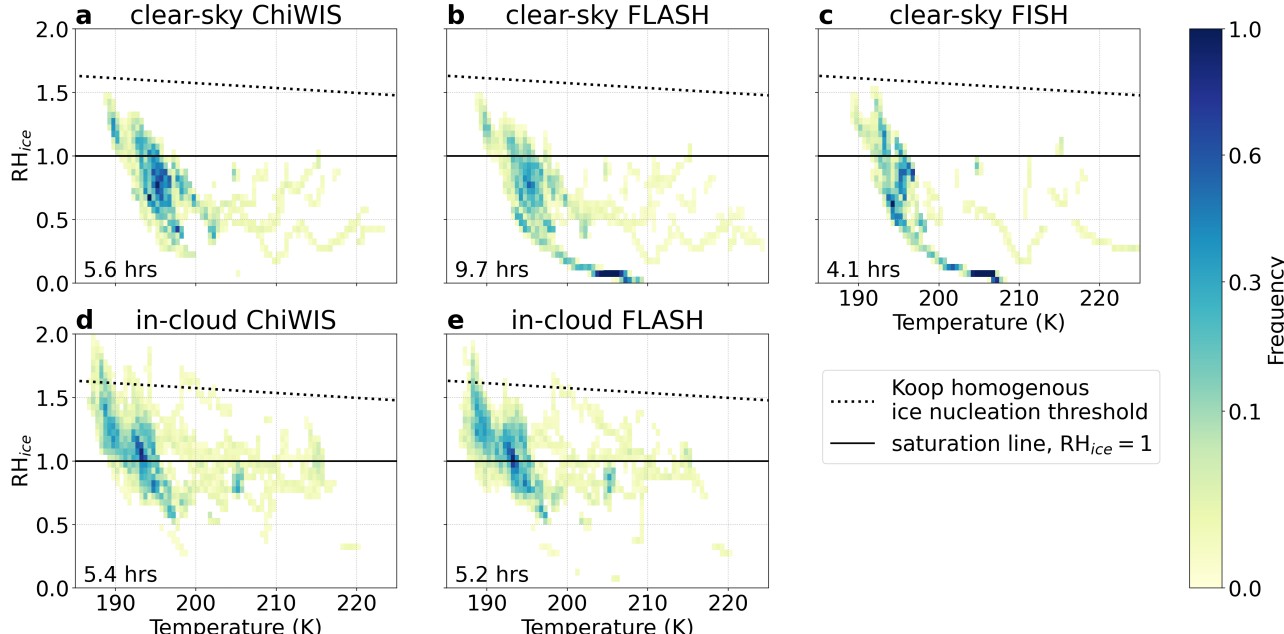

**Figure 7.** Relative humidity with respect to ice plotted against temperature for clear sky regions for a) ChiWIS, b) FLASH, and c) FISH, and for cloudy regions for d) ChiWIS and e) FLASH. The number of measurements in each 0.05 by 0.5 K bin is shown as a normalized frequency (so area under the histogram integrates to 1); total hours of measurements labelled in bottom left corner (Table 2). The dotted line shows the homogeneous nucleation threshold calculated according to Koop et al. (2000). Because the ChiWIS dataset excludes cases when cell pressure is out of regulation, panel a) does not include very low stratospheric $RH_{ice}$; see Fig. S8 for all data.

in-cloud ChiWIS compared to FLASH were $(-1.3 \pm 8.0)\%$ and 0.978, $(-1.4 \pm 6.2)\%$ and 0.958, and $(+0.3 \pm 7.7)\%$ and 0.887, respectively. The mean percentage difference for $RH_{ice}$ and $H_2O$ vapor have the same sign and magnitude for the three measurements. But unlike the $H_2O$ comparison where the correlation was stronger between ChiWIS and FLASH in-cloud compared to clear-sky, the opposite is true for the $RH_{ice}$ measurements. This is due to the spread in values from F8 with very lower $H_2O$ mixing ratios, but highly supersaturated $RH_{ice}$ values. Fig. S9 shows the same but for periods where ChiWIS cell pressure was out of regulation.

As with the $H_2O$ vapor measurements, more detailed information for each flight can be seen by time series and profile plots of $RH_{ice}$ (Fig. S3 and S4).

## 5 Mean atmospheric profile comparison

Lastly, we show that in situ measurements from the three aircraft hygrometers compare well with approximately co-located measurements from other sources. We compare mean profiles over the campaign with those from remote (MLS) and balloon-borne instruments (CFH). See Fig. 1 in Methods for spatial sampling by MLS. Both MLS v4 and v5 are shown. Fig. 9 shows

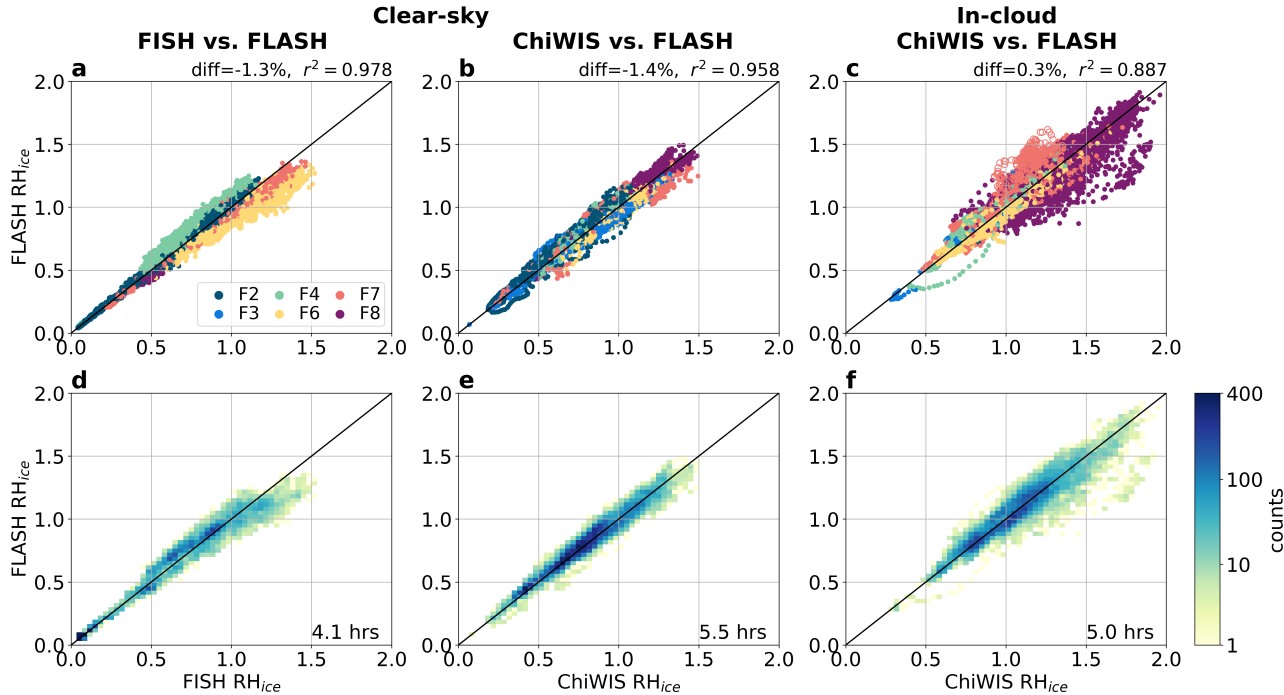

**Figure 8.** RH$_{ice}$ correlations between the three in situ hygrometers: a) clear-sky FISH vs. clear-sky FLASH, b) clear-sky ChiWIS vs. clear-sky FLASH, c) in-cloud ChiWIS vs. in-cloud FLASH, with the 1:1 line in black. Points in panels a)–c) are colored by flight number and plotted in random order. The total hours of paired observations is shown in the bottom right corner, and the percentage difference and $r^2$ coefficients are shown above each panel. Panels d)–f) show the same information as a)–c) but as the frequency of observations over all the flights in each 0.03 by 0.03 bin. The largest differences between paired observations occur on F8 in fresh convective outflows with very high in-cloud RH$_{ice}$. Flight-to-flight differences between FISH and FLASH H$_2$O are also noticeable in RH$_{ice}$. The open circles in panel c) on F7 mark the time period of disagreement between ChiWIS and FLASH as the airplane was ascending out of a deep dive.

mean profiles of all-sky H$_2$O with potential temperature as the vertical coordinate. We construct separate profiles for the two periods, to differentiate the warm/wet first half of the flight campaign and the cold/dry second half (Fig. 9a and b, respectively). Measurements from the balloons are only available starting on August 3, corresponding to the cold/dry period. To ensure that measurements are strictly comparable and restricted to gas-phase water vapor, we do not include FISH in this comparison

because it measures total water inside of clouds.

  The two periods show very similar profiles in the stratosphere above 405 K, but very different profiles in the TTL (also discussed in Khaykin et al. (2022)). During the warm/wet period (F2–F4), the mean H$_2$O profile is "L-shaped"; the background signal is monotonically decreasing from 360 to 440 K, with some sharp wet layers sampled by the aircraft up to 400 K. In contrast, the cold/dry period (F6–F8) has a non-monotonic H$_2$O profile with a hygropause coincident with the CPT around

350 382 K (dashed line on Fig. 9). Above the CPT there are wet layers up to around 405 K during this period, which was the maximum altitude of convective influence was observed by in situ measurements during the aircraft campaign (dotted line on

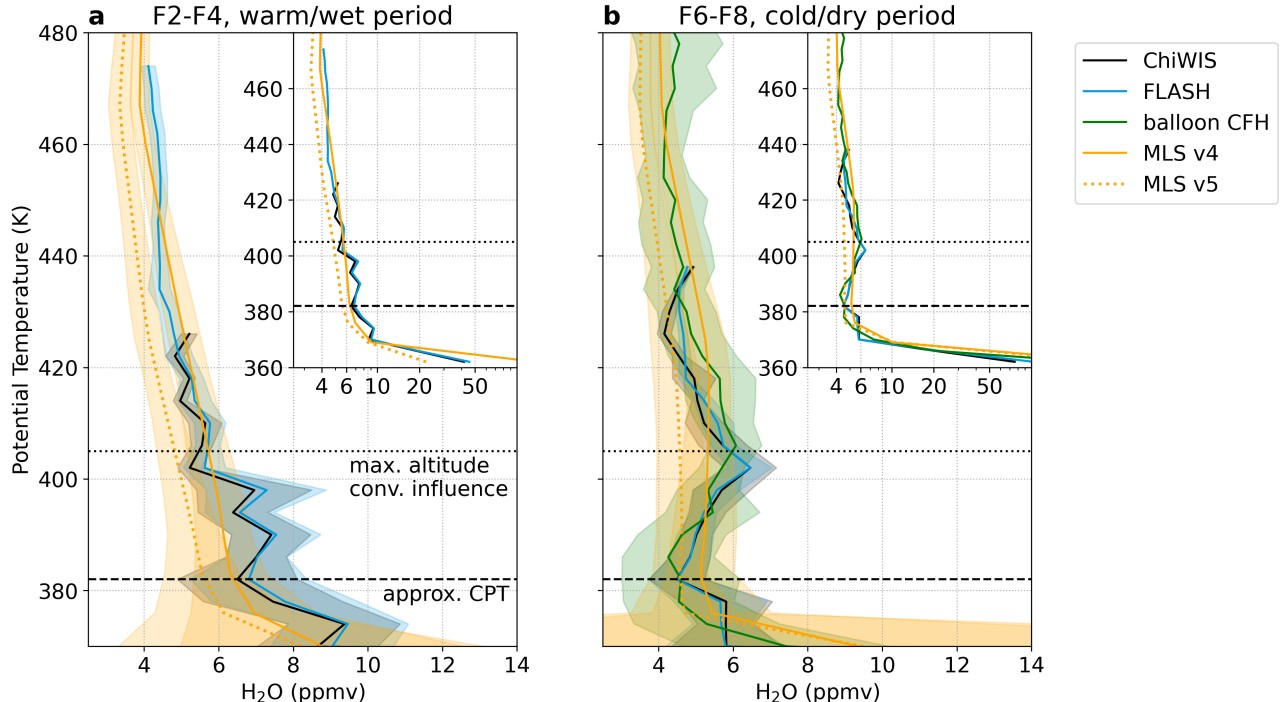

**Figure 9.** Mean all-sky water vapor profiles against potential temperature as measured by the aircraft hygrometers ChiWIS (black) and FLASH (blue), balloon-borne cryogenic frost point hygrometer (CFH, green), and the satellite microwave limb sounder (MLS, yellow) versions 4/5 (solid/dashed), with standard deviation shown as shaded region. Panel a) shows the warm/wet period of the campaign from 27 Jul to 3 Aug (F2–F4), b) shows the cold/dry period from 4 Aug to 10 Aug (F6–F8). FISH is not shown because it measures total water inside of clouds, while the other instruments measure only gas-phase water vapor. The CFH profile is constructed from 11 balloon soundings during the cold/dry period of the campaign and the MLS v5 (v4) profiles from 70 (65) retrievals in the warm/wet period and 56 (53) in cold/dry, over a region encompassing the flight tracks (shown in Fig. 1). Insets show mean profiles over a wider range of $H_2O$ mixing ratios, up to 100 ppmv. The heavy dashed line shows the approximate cold-point tropopause (CPT); in both period this coincides with the minimum measured water vapor. The dotted black line shows the approximate maximum level of convective influence. Water vapor profiles above this line are similar in both periods while the UTLS below varies dramatically. The MLS profiles still capture this temporal change despite their lower vertical resolution.

Fig. 9). The "L-shaped" profile from the warm/wet period is similar to the mean profile observed during the AMMA/SCOUT-O3 over West Africa discussed in Schiller et al. (2009). They trace back-trajectories of parcels observed over West Africa to convection in the ASM over India. Schiller et al. (2009) find this profile to be anomalous compared to the measurements from other tropical flight campaigns (SCOUT-O3 and TroCCiNOx), which had $H_2O$ profiles with a sharp hygropause at the CPT indicating efficient freeze-drying, similar to the profile observed during the StratoClim cold/dry period. It is notable that during StratoClim, diverse conditions of the monsoon were sampled, such that these two profiles shapes, previously only seen in different geographic regions, were observed during a single campaign.

Comparisons between these three platforms is challenging because the measurements were not perfectly coincident in space or time and the region sampled showed large day-to-day variability. The in situ measurements from the aircraft and balloons also have much higher spatial and temporal resolution than the MLS satellite instrument. Furthermore, sampling biases may be exacerbated in this comparison to do diurnal variations since the aircraft and MLS measured only during daytime while the balloons were only launched at dawn/dusk. Other sampling biases may be present such as cloud contamination, which to some extent all measurements are susceptible to. Overall MLS v4 shows a wet bias compared to v5 (of about 15% between 380–500 K), but both versions are able to discern trend across the campaign of a cooling/drying of the UTLS seen by the aircraft measurements. Small-scale vertical structures caused by convection (Khaykin et al., 2022) are smeared out due to the coarse resolution of MLS. Below 380 K the sampling is quite sparse and the MLS measurements show very large variance.

Despite these worries, the measurements show good agreement between 380 and 440 K. During the warm/wet period, MLS v5 shows a significant dry bias compared to the aircraft instruments of $(-19 \pm 7)\%$ and $(-22 \pm 6)\%$ for ChiWIS and FLASH, respectively. During the cold/dry period, MLS v5 shows an insignificant dry bias of $(-12 \pm 12)\%$, $(-11 \pm 12)\%$, and $(-5 \pm 15)\%$ compared to ChiWIS, FLASH, and the balloon CFH, respectively. Because MLS v4 is 15% wetter compared to v5 in this altitude range, MLS v4 actually agrees more closely with the in situ measurements, reporting no statistically significant differences with any of the instruments during either period of the campaign.

## 6   Summary and Conclusions

In this study, we intercompare water vapor measurements from the Asian summer monsoon UTLS made by three in situ hygrometers on board the M-55 Geophysica aircraft during the StratoClim campaign. This campaign constituted the inaugural flights for the ChiWIS spectrometer which we compared with the established Lyman-$\alpha$ hygrometers FLASH and FISH. We also validate the airborne instruments by comparing to an in situ balloon-borne hygrometer and the MLS satellite instrument. We show excellent agreement between the in situ instruments; mean differences between paired observations at stratospheric mixing ratios ($H_2O <$ 10 ppmv) were $(-1.5 \pm 8.0)\%$ for clear-sky FISH and $(-0.4 \pm 6.8)\%$ for all-sky ChiWIS compared to FLASH, respectively. For mixing ratios up to 100 ppmv the mean differences were $(-1.3 \pm 8.0)\%$ and $(-0.6 \pm 7.0)\%$, respectively. Comparisons of $RH_{ice}$ further validated the instrument performances; differences in $RH_{ice}$ measurements were less than $\pm 1\%$. Mean $RH_{ice}$ values in clear-sky were 0.51–0.52 and in cloudy conditions were 1.05–1.07 with very few in-cloud measurements above the homogeneous nucleation threshold from Koop et al. (2000) at temperatures $<$ 190 K. Campaign-mean profiles of UTLS $H_2O$ from the airborne in situ hygrometers agreed with the balloon CFH and remotely-sensed MLS profiles to within 5%.

The agreement between the in situ water vapor measurements (ChiWIS and FLASH) is remarkably good (mean percentage differences of $1 - 5\%$ for each flight) and exceeds that found by previous intercomparison studies ($2.5 - 30\%$) (Vömel et al., 2007; Weinstock et al., 2009; Rollins et al., 2014; Meyer et al., 2015; Kaufmann et al., 2018). This agreement is the result of fastidious engineering before the flight campaign and robust communication between instrument teams during subsequent data analysis. For the new ChiWIS instrument, extra care was taken during construction to prevent, as much as possible, adsorption

and desorption of water vapor in the inlet plumbing. We also benefited from improved spectroscopic data in the HITRAN2016 database (Gordon et al., 2017) that reduced the line strength uncertainty from $\approx 10\%$ down to only $2 - 3\%$. Finally, frequent communication between instrument teams and a preliminary intercomparison effort led to the early discovery of measurement problems from both the FLASH and ChiWIS instruments. The instruments did not calibrate relative to each other, but rather these issues were corrected for independently based on laboratory experiments and calibration runs with gas standards. More detail on the ChiWIS laser "pedestal" (stray light) that was corrected for with calibration based on laboratory experiments can be found in Clouser et al. (2022, in prep.). These consistent in situ measurements of UTLS water vapor represent progress in the field to improve the accuracy of instrumentation in this difficult measurement regime.

The agreement found here allows for unambiguous scientific interpretation and detailed process-level studies that will clarify the role of the ASM on the global UTLS water vapor budget. During StratoClim, diverse conditions of the monsoon were sampled in a single campaign, as noted by the two characteristic profile shapes shown in Figure 9, which were previously only seen in different geographic regions (Schiller et al., 2009). See Khaykin et al. (2022) for an analysis of the dual role of convection in hydration and dehydration. Confidence in this analysis is made possible by this rigorous intercomparison effort. Furthermore, this intercomparison serves as robust validation for the new ChiWIS flight instrument which has made the first measurements of water isotopologues in the ASM UTLS. Finally, the extreme conditions sampled during the StratoClim campaign, especially the very cold and dry conditions of the later flights can provide rare in situ data to check new theories of homogeneous nucleation and cirrus cloud formation.

*Data availability.* Water vapor measurements (ChiWIS, FLASH, and FISH), met data from UCSE and TDC, and MAS backscatter measurements from the StratoClim aircraft campaign are available on the HALO database at https://halo-db.pa.op.dlr.de/mission/101 (German Aerospace Center, 2021). Particle measurements from NIXE-CAPS are available in Krämer et al. (2020a). Balloon data are available in the supplement of Brunamonti et al. (2019). MLS data are publicly available at http://disc.sci.gsfc.nasa.gov/Aura/data-holdings/MLS. Analysis and plotting scripts for this paper are available at https://github.com/claresinger/StratoClim_H2O_Intercomparison.

*Author contributions.* BWC, CES, and EJM performed and processed the ChiWIS measurements; SMK and AL performed and processed the FLASH measurements; CR, NS, AA, and MK performed and processed the FISH and NIXE-CAPS measurements; SB and TP performed and processed the balloonsonde measurements; FC performed and processed the MAS measurements; SMK retrieved and processed the MLS measurements. CES performed the intercomparison analysis, prepared the figures, and wrote the manuscript. All authors commented on the manuscript.

*Competing interests.* The authors declare that they have no conflict of interest.

*Acknowledgements.* This research has been supported by the European Commission, Seventh Framework Programme (StratoClim, Grant #603557). CES was supported by a UChicago Summer Undergraduate Research Fellowship and NSF Grant #1743753 to EJM. We thank the three anonymous referees for their constructive remarks.

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
