# Peer review of "Intercomparison of upper tropospheric and lower stratospheric water vapor measurements over the Asian Summer Monsoon during the StratoClim Campaign"

_Atmospheric Measurement Techniques, 2022_

## Author Comment (AC1)

**Response to anonymous reviewer #1**

Thank you very much for your positive review and your helpful comments – they have improved the manuscript greatly.

**Major comments:**

The study uses v4.3 for the comparison with MLS water vapor measurements. The MLS team has already released v5.0, which addresses some significant biases and drifts that do affect the comparison with the in situ measurements. I would strongly suggest using v5 instead of v4.3 as the results would look slightly differently.
→ This was commented on by all reviewers and the following reply has been sent to all reviewers.

Thank you for alerting the authors to the newest version of MLS. At the time this analysis was started v5 had not yet been released. We have updated the analysis to use MLS v5. The results are similar, but in the UTLS (~68 hPa), v5 is about 15% drier than v4 across the whole globe (not shown). In consultation with the MLS team at JPL, there is not a known cause of the larger than expected (from changing sideband fractions) discrepancy at these levels between the two versions, which is about twice as large as reported by Livesey et al. (2021) in section 4: https://doi.org/10.5194/acp-21-15409-2021/. For completeness we have included both v4 and v5 in Fig 9 (previously Fig. 8). Now MLS appears drier than in situ measurements (aircraft and balloon) in both the warm/wet and cold/dry halves of the campaign, but still captures the qualitative shift of drying through time observed by the aircraft instruments across these two periods. We have added a short discussion about the differences between MLS v4 and v5 to the text:

"Here we use 126 water vapor profiles spatially and temporally co-located with the StratoClim flights as a point of comparison (shown in Fig. 1a). We use version 5.0 (v5) profiles which were selected in the region between 20–30°N and 78–92°E during the campaign dates of 27 July – 10 August 2017, using screening criteria from Livesey et al. (2022). We also show MLS version 4.3 (v4) profiles (only 118) which were selected using screening criteria in Livesey et al. (2020). We interpolate the $H_2O$ profiles onto a potential temperature grid using the MLS temperature product provided at the same pressure levels. MLS v5 includes a correction on the $H_2O$ retrievals described in Livesey et al. (2021), which results in an approximately spatially uniform drying at 68 hPa of about 15%.
…
Overall MLS v4 shows a wet bias compared to v5 (of about 15% between 380-500 K), but both versions are able to discern trend across the campaign of a cooling/drying of the UTLS seen by the aircraft measurements.
…
During the warm/wet period, MLS v5 shows a significant dry bias compared to the aircraft instruments of (-19 ± 7)% and (-22 ± 6)% for ChiWIS and FLASH, respectively.

During the cold/dry period, MLS v5 shows an insignificant dry bias of (-12 ± 12)%, (-11 ± 12)%, and (-5 ± 15)% compared to ChiWIS, FLASH, and the balloon CFH, respectively. Because MLS v4 is 15% wetter compared to v5 in this altitude range, MLS v4 actually agrees more closely with the in situ measurements, reporting no statistically significant differences with any of the instruments during either period of the campaign."

**Minor comments:**

I would suggest moving Figure S6 from the supplement to the main document. It is quite helpful in understanding the discussion of the cloud determination.
→ Thank you, Figure S6 has been moved into the main text.

Since the authors refer to Figure S9 twice, it too could possibly be moved to the main text.
→ Figure S9 is shown to present the reader with a fuller picture of the atmospheric state, including data from ChiWIS that was excluded from the intercomparison. So as not to confuse the reader on which datapoints are being intercompared, we have kept S9 as a supplemental figure. Thank you.

Lines 246ff: It would be quite useful to have an overview table, which lists for each flight the total flight hours, total number of data points, total number of measurements in cloud and out of cloud. Without this, the number of data points given here miss perspective.
→ A new table (Table 2) has been added with a summary of the flight hours for the 6 flights discussed in this paper.

The comparisons in Figure 8 look quite good. Nevertheless, there may be some sampling biases in this comparison, since the balloon-borne measurements are typically launched only in non-precipitating conditions, and MLS is sensitive to cloud contamination in the UTLS. A short discussion about how this potential sampling bias may influence the comparison could be helpful.
→ Thanks for this suggestion. There are a variety of possible sampling biases that may be present in this comparison. This makes it all the more impressive that there is such good agreement across platforms. We have added the following discussion about Fig 9 (previously 8):

"Comparisons between these three platforms is challenging because the measurements were not perfectly coincident in space or time and the region sampled showed large day-to-day variability. The in situ measurements from the aircraft and balloons also have much higher spatial and temporal resolution than the MLS satellite instrument. Furthermore, sampling biases may be exacerbated in this comparison to do diurnal variations since the aircraft and MLS measured only during daytime while the balloons were only launched at dawn/dusk. Other sampling biases may be present such as cloud contamination, which to some extent all measurements are susceptible to."

Lines 374: Not unexpected, the comparison was not blind and there have been many interactions between the different teams during the campaign. I have confidence in each team,

but it would still be good to know to what extent if any instrument calibrations were adjusted during the campaign based on input from the other teams.

→ Thank you for this comment. We appreciate now that the sentence as written may be misleading. We did not mean that any instrument calibrations were adjusted based on other teams' data, but rather that problems were initially discovered based on a preliminary instrument intercomparison and then confirmed with laboratory experiments. Calibrations were applied based solely on those laboratory experiments and calibration runs with gas standards. We have updated the text to clarify this as follows:

"Finally, frequent communication between instrument teams and a preliminary intercomparison effort led to the early discovery of measurement problems from both the FLASH and ChiWIS instruments. The instruments did not calibrate relative to each other, but rather these issues corrected for independently based on laboratory experiments and calibration runs with gas standards. More detail on the ChiWIS laser "pedestal" (stray light) that was corrected for with calibration based on laboratory experiments can be found in Clouser et al. (2022, in prep.)."

**Technical comments:**

Line 150: delete the stray comma.
→ Done.

Lines 269f: Better write: "During the stair stepping, there were two moist layers around …"
→ Done.

In Figure 5: I would suggest making the vertical axes for potential temperature and H2O the same between the upper and lower group of panels.
→ Thank you for this suggestion. While we agree that it is convenient to use the same vertical axes on subplots, due to the fact that different flights sampled different altitude ranges, doing so makes the F7 data more difficult to see because it is squashed together. We have chosen to leave Figure 5 as is, but included a statement in the caption that points out the different vertical axes to the reader.

---

## Author Comment (AC2)

**Response to anonymous reviewer #2**

We thank the reviewer for their very helpful suggestions and their positive review.

1) The abstract states " In clear-sky UTLS conditions (H2O < 10 ppmv), mean differences between ChiWIS and FLASH were only -1.42% and those between FISH and FLASH only -1.47%. Agreement between ChiWIS and FLASH for in-cloud conditions is even tighter, at +0.74%. In general, ChiWIS and FLASH agreed to better than 10% for 92% (87%) of clear-sky (in-cloud) datapoints." I'm a bit confused between the order 1% values noted in the first sentence, and the 10% value noted in the second. Is the second sentence valid for cases where H2O is larger than 10ppmv? Perhaps instead, the uncertainty needs to be included in the first sentence (ie, what's the Standard deviation on the -1.42% for ChiWIS and FLASH?) Please made this clear. Also, include the range in the statement on line 253 on page 10.

→ We received a similar comment from another reviewer about the confusing nature of these two sentences in the abstract. The following comment has been sent to both reviewers:

The goal of the first sentence was to describe the mean differences between instruments across the whole campaign. The goal of the second sentence was to describe the variability of the differences in simultaneous measurements, essentially providing the same information as a standard deviation to the mean values would, but the authors thought this format of 'what fraction of simultaneous measurements agreed better than 10%?' was more intuitive. Furthermore, this way of presenting the statistics aligned nicely with the dotted and dashed lines shown in Fig 3 (previously Fig 2).

However, clearly this phrasing has created more confusion rather than helping. We have chosen to remove this second sentence and instead have added the standard deviations (in addition to mean) for instrument differences. This information is also now presented in Table 3, as suggested by the reviewers. Thank you!

We have also added the standard deviations shown in (now) Fig 5 to the text as suggested. Thank you.

2) Figures in supplement: It would be easier to look at on a laptop screen if you include the values on the horizontal axis for figure s2 and s4
→ X-axis labels for all subplots have been added to Figures S2 and S4. Thank you.

3) Re: MLS, the most recent retrieval is version 5, which at least partially corrected for an instrumental drift. It would be worthwhile redoing the comparisons with the new retrieval.
→ This was commented on by all reviewers and the following reply has been sent to all reviewers.

Thank you for alerting the authors to the newest version of MLS. At the time this analysis was started v5 had not yet been released. We have updated the analysis to use MLS v5. The results are similar, but in the UTLS (~68 hPa), v5 is about 15% drier than v4 across the whole globe (not

shown). In consultation with the MLS team at JPL, there is not a known cause of the larger than expected (from changing sideband fractions) discrepancy at these levels between the two versions, which is about twice as large as reported by Livesey et al. (2021) in section 4: https://doi.org/10.5194/acp-21-15409-2021/. For completeness we have included both v4 and v5 in Fig 9 (previously Fig. 8). Now MLS appears drier than in situ measurements (aircraft and balloon) in both the warm/wet and cold/dry halves of the campaign, but still captures the qualitative shift of drying through time observed by the aircraft instruments across these two periods. We have added a short discussion about the differences between MLS v4 and v5 to the text:

"Here we use 126 water vapor profiles spatially and temporally co-located with the StratoClim flights as a point of comparison (shown in Fig. 1a). We use version 5.0 (v5) profiles which were selected in the region between 20–30°N and 78–92°E during the campaign dates of 27 July – 10 August 2017, using screening criteria from Livesey et al. (2022). We also show MLS version 4.3 (v4) profiles (only 118) which were selected using screening criteria in Livesey et al. (2020). We interpolate the H2O profiles onto a potential temperature grid using the MLS temperature product provided at the same pressure levels. MLS v5 includes a correction on the H2O retrievals described in Livesey et al. (2021), which results in an approximately spatially uniform drying at 68 hPa of about 15%.

…

Overall MLS v4 shows a wet bias compared to v5 (of about 15% between 380-500 K), but both versions are able to discern trend across the campaign of a cooling/drying of the UTLS seen by the aircraft measurements.

…

During the warm/wet period, MLS v5 shows a significant dry bias compared to the aircraft instruments of (-19 ± 7)% and (-22 ± 6)% for ChiWIS and FLASH, respectively.
During the cold/dry period, MLS v5 shows an insignificant dry bias of (-12 ± 12)%, (-11 ± 12)%, and (-5 ± 15)% compared to ChiWIS, FLASH, and the balloon CFH, respectively. Because MLS v4 is 15% wetter compared to v5 in this altitude range, MLS v4 actually agrees more closely with the in situ measurements, reporting no statistically significant differences with any of the instruments during either period of the campaign."

---

## Author Comment (AC3)

**Response to anonymous reviewer #3**

Thank you very much for this detailed review.

**Major comments:**

The near-zero bias values between instruments over the full campaign are misleading because a much larger positive bias for one flight effectively "cancels out" a similar magnitude negative bias for another flight. For example, for FISH-FLASH biases for F4 and F6 in Figure 3. F4 shows a negative bias of 10-15% and F6 shows positive bias of 10-15% for F6. When combined over the entire campaign, the two biases basically add to zero. I would like to suggest that summary bias statistics for each instrument pair, for each flight, including uncertainties, be provided in a table. Then the reader can see not only the large biases during specific flights, but also how these biases change in both magnitude and sign from one flight to the next. Yes, this information can be gleaned from Figure 3, but having the statistics readily available in a table will make them much more accessible.
→ Thank you for this suggestion. The authors agree a table helps make this information more accessible and makes the flight-to-flight differences more apparent. A new table (Table 3) has been added to the paper which includes mean and standard deviation of percentage differences between paired instruments for each flight, each of the two periods (warm/wet and cold/dry), and the campaign overall. Differences between FISH and ChiWIS are included here for completeness (but left out of figures to avoid overloading on too much repetitive information). Differences are also broken down between clear-sky, cloudy, and all-sky for ChiWIS vs. FLASH.

I'm sure others have pointed out that the MLS4.3 data used in the paper have now been replaced by v5 that reduces some of the "wet" biases that had crept into the older v4 retrievals. Although I don't find it essential to replace the MLSv4 data in the paper (e.g., profile statistics in Figure 8), it should be stated that similar profile statistics based on MLSv5 data would generally shift them to lower mixing ratios by 0.2-0.3 ppm. The "dry bias" mentioned in L348 would become even greater, while the "wet bias" (L349) would be reduced by v5. The importance of this MLS data to the comparison with the aircraft instruments would become much greater if some quantitative "coincidence" criteria with the aircraft flights were mentioned. How far apart in time and space were these MLS and Geophysica measurements made?
→ This was commented on by all reviewers and the following reply has been sent to all reviewers.

Thank you for alerting the authors to the newest version of MLS. At the time this analysis was started v5 had not yet been released. We have updated the analysis to use MLS v5. The results are similar, but in the UTLS (~68 hPa), v5 is about 15% drier than v4 across the whole globe (not shown). In consultation with the MLS team at JPL, there is not a known cause of the larger than expected (from changing sideband fractions) discrepancy at these levels between the two versions, which is about twice as large as reported by Livesey et al. (2021) in section 4: https://doi.org/10.5194/acp-21-15409-2021/. For completeness we have included both v4 and

v5 in Fig 9 (previously Fig. 8). Now MLS appears drier than in situ measurements (aircraft and balloon) in both the warm/wet and cold/dry halves of the campaign, but still captures the qualitative shift of drying through time observed by the aircraft instruments across these two periods. We have added a short discussion about the differences between MLS v4 and v5 to the text:

"Here we use 126 water vapor profiles spatially and temporally co-located with the StratoClim flights as a point of comparison (shown in Fig. 1a). We use version 5.0 (v5) profiles which were selected in the region between 20–30°N and 78–92°E during the campaign dates of 27 July – 10 August 2017, using screening criteria from Livesey et al. (2022). We also show MLS version 4.3 (v4) profiles (only 118) which were selected using screening criteria in Livesey et al. (2020). We interpolate the H2O profiles onto a potential temperature grid using the MLS temperature product provided at the same pressure levels. MLS v5 includes a correction on the H2O retrievals described in Livesey et al. (2021), which results in an approximately spatially uniform drying at 68 hPa of about 15%.
…
Overall MLS v4 shows a wet bias compared to v5 (of about 15% between 380-500 K), but both versions are able to discern trend across the campaign of a cooling/drying of the UTLS seen by the aircraft measurements.
…
During the warm/wet period, MLS v5 shows a significant dry bias compared to the aircraft instruments of (-19 ± 7)% and (-22 ± 6)% for ChiWIS and FLASH, respectively.
During the cold/dry period, MLS v5 shows an insignificant dry bias of (-12 ± 12)%, (-11 ± 12)%, and (-5 ± 15)% compared to ChiWIS, FLASH, and the balloon CFH, respectively. Because MLS v4 is 15% wetter compared to v5 in this altitude range, MLS v4 actually agrees more closely with the in situ measurements, reporting no statistically significant differences with any of the instruments during either period of the campaign."

I am somewhat baffled by the treatment (and lack of use) of the 11 different CFH profiles that were obtained during the campaign. Was the only use of this data to consolidate 11 individual profiles down to one "statistical" (mean ± stddev) profile to compare with the aircraft data? Were the individual CFH profiles of little use in comparing to the aircraft data for individual flights? For example, would CFH profiles in coincidence with F4 and F6 help explain why the bias between FISH and FLASH changed sign between the two flights? If there is a good reason why the CFH profiles are used only to create one "statistical" profile it would be beneficial to include such a statement, otherwise the reader is left wondering why the balloon profiles play so little role in the intercomparison.
→ Thank you for this comment. We have added some clarification to the text that explains why individual balloon soundings should not be compared to individual aircraft flights. Essentially it boils down to the fact that the soundings were not synchronous with the flights -- for logistical reasons the balloons were only launched at dusk/dawn and the aircraft flights were only during the day. Furthermore, the balloon soundings did not necessarily take place on the same date as the flights. And thus, because of the large day-to-day variability and diurnal variability in WV,

there is no guarantee the balloon measurements would match the aircraft even on average, but certainly not for individual flights. Additionally, the balloons were not launched from the airport, and some of the aircraft flights went much further away from Kathmandu than the balloons traveled, so any spatial variability also makes a direct comparison challenging. We include the comparison in the final figure (Fig 9, previously Fig 8) for completeness and in fact find fairly good agreement. The following discussion has been added about the sampling biases that may be present in this bulk comparison across different measurement platforms:

"Comparisons between these three platforms is challenging because the measurements were not perfectly coincident in space or time and the region sampled showed large day-to-day variability. The in situ measurements from the aircraft and balloons also have much higher spatial and temporal resolution than the MLS satellite instrument. Furthermore, sampling biases may be exacerbated in this comparison to do diurnal variations since the aircraft and MLS measured only during daytime while the balloons were only launched at dawn/dusk. Other sampling biases may be present such as cloud contamination, which to some extent all measurements are susceptible to."

**Minor comments:**

Lines 10-11 vs line 12: This confused me when I first read the abstract. Overall mean differences are on the order of 1%, but then "In general, CWIS and FLASH agreed to better that 10% for 92% ...". I thought there was probably a typographical error. However, after reading the paper, I now understand why the overall 1% bias values are so small - the topic of my first significant concern above. I suggest re-working the abstract text describing overall (full campaign) bias statistics to be less misleading, as I think full-campaign statistics are not very meaningful when there are large flight-to-flight differences in the biases.

L10-11: Need to include uncertainty values for mean differences
→ Responding to the two above comments: We received a similar comment from another reviewer about the confusing nature of these two sentences in the abstract. The following comment has been sent to both reviewers:

The goal of the first sentence was to describe the mean differences between instruments across the whole campaign. The goal of the second sentence was to describe the variability of the differences in simultaneous measurements, essentially providing the same information as a standard deviation to the mean values would, but the authors thought this format of 'what fraction of simultaneous measurements agreed better than 10%?' was more intuitive. Furthermore, this way of presenting the statistics aligned nicely with the dotted and dashed lines shown in Fig 3 (previously Fig 2).

However, clearly this phrasing has created more confusion rather than helping. We have chosen to remove this second sentence and instead have added the standard deviations (in

addition to mean) for instrument differences. This information is also now presented in Table 3, as suggested by the reviewers. Thank you!

L13: change "to" to "within"
→ Done.

L15: The detection of "fine-scale spatial structures" depends not only on high precision (not accuracy), but also on instrument response time. This sentence is a bit of "grandstanding" relative to the contents of this paper.
→ "Accuracy" here is required for quantifying the role of the ASM on the stratospheric water vapor budget. Precision (and you are correct, time resolution) is needed for fine-scale spatial structures. Of course, what is meant by "fine" is not defined here, but in our observations, it was precision, not temporal resolution, that limited the detection of smaller-scale water vapor variations.

L18: With mixing ratios > 10,000 ppm at Earth's surface, water vapor is not a "trace gas"
→ At the surface water vapor can be up to a few %, but in the UTLS region at < 10 ppmv, it is a trace component. But since it is not necessary for the meaning of this sentence, we have removed the word "trace" and settled that "water vapor is one of the most important gases" in total.

L20: "that roughly doubles the anthropogenic warming from carbon dioxide alone" is a very mis-interpretable statement without additional context.
→ We have carefully checked the reference supplied at the end of this sentence (to Dessler, 2008) to ensure that it provides the necessary context about how the water vapor feedback amplifies direct warming from CO2. In the first sentence of the introduction of the cited paper the authors say: "The water-vapor feedback is one of the most important in our climate system, with the capacity to about double the direct warming from greenhouse gas increases [Manabe and Wetherald, 1967; Randall et al., 2007]." The rest of the text goes on to explain the satellite measurements of water vapor from AIRS and quantify the magnitude of the water vapor feedback to justify this qualitative statement.

L26: Mole fractions are not "concentrations".
→ Thank you for spotting this colloquial usage of the term concentrations. We have replaced it with "amounts."

L36: Is water vapor a "pollutant" in Earth's atmosphere?
→ While stratospheric H2O has harmful consequences, such as ozone destruction, we recognize that calling it a "pollutant" may be confusing and have changed the word "including" to "and." Thank you.

L42: "low" is such a relative adjective. Most trace gases exist at mixing ratios <5 ppm, some < 5 ppt, so why is 5 ppm considered "low" for water vapor? It is because water vapor can also be found in the atmosphere at >10,000 ppm. Anytime an instrument is carried from the very wet

lower troposphere to the UTLS it is challenge to measure the very dry air there. That's why at 5 ppm the water vapor mixing ratio is low.

→ Yes, exactly. Contamination during ascent is a huge engineering challenge.

L46: I would change "Although" for "Because"

→ Thank you for this suggestion. "Although" is used here to say that just because the problem has been recognized for a long time, it is still a problem and these studies are still necessary.

L55: NOAA frost point hygrometer is also known as the NOAA FPH.

→ We tried to avoid acronyms for terms which aren't repeated extensively in the paper.

L64: Voemel and Hall published what can be described as "detailed uncertainty analysis of water vapor measurements by the CFH and FPH, respectively," rather than "an intercomparison"

→ Thank you. The term intercomparison has been removed.

L74: remove "with" before "within"

→ Thank you for catching this typo.

L81: Knowing that none of these three instrument directly measures RH, how can RH values provide accuracy information when the RH uncertainties are a combination of the uncertainties of simultaneous WV and T measurements?

→ The reviewer is correct that RH uncertainties come from both T and WV uncertanties, and something we discuss later in the paper. The reason for analyzing RH, however, is that there are robust physical processes controlling the magnitude of RH that do not exist for T or WV. E.g., the WV could be 4ppmv or 8ppmv and neither of these values are at all meaningful in terms of an accuracy. However, once we put them into context with the temperature and pressure, and calculate RH, we can suddenly make statements like 200% RH is far too high. While this does not allow us to determine with absolute certainty that the WV is biased high vs. the T is biased low, it is one way we can put constraints on absolute accuracy. This is discussed in more detail in section 4 and Fig. 7 (previously 6).

L83: the second occurrence of "in situ measurements" can be omitted

→ Thank you. Done.

L90: There have been previous in situ measurements in the ASM. For example, see https://doi.org/10.1016/j.atmosres.2022.106093

→ Thank you for directing us to this new paper. We have clarified this sentence to read "the first in situ measurements from aircraft of the ASM" and added a citation to Ma et al. (2022) as well as Bian et al. (2012) and Vernier et al. (2018) which describe other balloon measurements of the ASM.

L219: "finite" is extremely vague here. Is the time needed on the order of 10s of seconds? 100s of seconds? Please be more quantitative here.

→ It is on the order of 10 seconds. Precise quantitative information is contained in Fig. S6 (previously S7). We have added this order of magnitude in parentheticals to the paper.

L236: "of better 1.5%" makes no sense.
→ Missing word added: "is better than" – thank you for noticing this typo.

Figure 3: This figure intrigues me because there are no tails on both sides of the distributions. Do the normalized PDFs really drop from viewable values to unviewable values in these panels? Why does the y-scale go from 10^-3 to 10^-5 in one increment when all the other increments are a single factor of 10?
→ This figure is correct. The y-scale increments are each spaced 10^2 apart. The tails drop off precipitously because the histograms are plotted on a logarithmic y-scale.

L253: These mean differences (and every single one in the paper) should be accompanied by a uncertainty value, most like two standard deviations of the mean.
→ Thank you for the suggestion. Standard deviations have been added to these values (as already shown on the figure itself).

Figure 4: Yellow is never a good choice for colored markers in Figures. It is difficult to tell where the yellow markers begin and end in this figure. The terminology "heat map" is new to me. What does it mean?
→ A "heat map" is synonymous with a 2D histogram. We are showing the number of observations in each 2D bin of % diff and potential temperature. The color scale was purposefully chosen to fade to very light colors where there are very few observations.

L282: You don't need pressure to calculate RH, only WV partial pressure and ambient temperature.
→ Given the mixing ratio of water vapor, we need to know the ambient temperature and ambient pressure to calculate RH.

Figures 6, 7: I understand the purpose of showing values of RH-ice here against the theoretical limit to assess WV measurement accuracy, but with anything but the highest accuracy T measurements used to derive RH values, calculated RH values have uncertainty contributions from the T measurements as well as the WV measurements. If RH-ice slightly exceeds the theoretical limit, is the WV accuracy to blame? or the T accuracy? or both?
→ Yes, of course both uncertainties in T and WV are contributing to uncertainties in RH and we cannot say with any certainty that these RHi values above the theoretical limit are due to WV. This is already discussed in the paper (previously, L154 and L305).

L154: "Estimated accuracy and precision are 0.5 K and 0.1 K, respectively, and dominate uncertainty in relative humidity. The measurement uncertainty on temperature alone (assuming a temperature of 200 K and a perfect measurement of H2O and pressure) translates to a fractional uncertainty ($\Delta$RHice / RHice) of about 0.08. Conversely, a measurement uncertainty from H2O alone would need to be as large as 0.4 ppmv at a background

stratospheric value of 5 ppmv to produce the same fractional uncertainty in derived relative humidity."

**L305:** "Both ChiWIS and FLASH report very infrequent, but non-zero, measured points above the homogeneous nucleation thresh- old at very low temperatures (T < 190 K) during in-cloud periods. This may be attributed to measurement uncertainty on either the H2O or temperature values, both of which are very difficult to measure at these low mixing ratios and cold temperatures."

What I do not understand is the point-by-point comparison of RH-ice values from the three hygrometers. How does this analysis shed any more (or different) light on the previous comparison of WV mixing ratios? RH values are a derived quantity, while WV mixing ratios are the direct measurements. What does Figure 8 add that Figure 2 hasn't already shown?
→ The reviewer is correct that this is not adding new information per se, but rather comparing point-by-point RHi measurements is just a different way to visualize the comparison between instruments. However, we still believe it is a useful way to compare instruments because Rhi highlights different WV measurements than just looking at absolute magnitudes. For instance, one important difference that arises when looking in Rhi space rather than WV space directly is the in-cloud comparison between FLASH and ChiWIS: for WV the instruments have very similar agreement during clear-sky and in-cloud times (r^2 = 0.928 and 0.930, respectively), while for RHi the agreement during clear-sky is significantly better than in-cloud (r^2 = 0.958 and 0.887, respectively). This difference arises from F8 measurements where the lowest absolute values of WV were observed, so when we look in WV these points all cluster at the small end and small variations are harder to see, but when we transform into RHi space where the RHi is very large (1.5+) then these small differences are magnified. These ideas are already discussed in (previously) L317-319.

L368: This sentence is far too "grandstanding" for what is discussed in this paper. How can the attribution of "good agreement" for the full campaign data, knowing that individual flight biases are high in some cases, and flip from negative to positive from one flight to the next, be the product of these two things? Please tone this sentence down as it far too overreaching.
→ We have updated this sentence to explicitly consider the two water vapor instruments (ChiWIS and FLASH) which showed considerably less flight-to-flight variations. The new Table 3 is helpful (thank you for this suggestion) to see these variations directly and to see the quality of the agreement between the two vapor hygrometers. For all flights the differences were less than 5% (3.5% excluding F2). Discussion of how flight-to-flight variations allow instrument disagreements to cancel out has been added (copied below):

"When examining the performance of FISH compared to FLASH, flight-to-flight variations tend to cancel out when averaged across the campaign: in Table 3 (first row), the mean difference during the first "warm/wet" half of the campaign compared to the second "cold/dry" half were −4.4% and +9.6%, which averaged together meant only a -1.5% mean difference across the whole campaign, but with a sizable variance (±8.0%)."

It also makes it sound like you all colluded after the flights to make sure the data from different instruments agree with one another. It is clear these data sets were not prepared or submitted under "blind" conditions, but "robust communication between instrument teams during subsequent data analysis" probably makes it sounds more conspiratorial than it was.

→ Thank you for this comment. We appreciate now that the sentence as written may be misleading. We did not mean that any instrument calibrations were adjusted based on other teams' data, but rather that problems were initially discovered based on a preliminary instrument intercomparison and then confirmed with laboratory experiments and calibrations were applied based solely on those laboratory experiments and calibration runs with gas standards. We have updated the text to clarify this as follows:

"Finally, frequent communication between instrument teams and a preliminary intercomparison effort led to the early discovery of measurement problems from both the FLASH and ChiWIS instruments. The instruments did not calibrate relative to each other, but rather these issues corrected for independently based on laboratory experiments and calibration runs with gas standards. More detail on the ChiWIS laser "pedestal" (stray light) that was corrected for with calibration based on laboratory experiments can be found in Clouser et al. (2022, in prep.)."

L374: I think "major progress" overreaches the findings of this comparison paper. Some instrument pairs were found to have large (10-15%) mean biases during some flights. That's not new. The biases sometime changed sign from one flight to the next. Definitely not new. When all the flight are combined, the large biases observed for some individual flights were diminished by similar biases of opposite sign for other flights. This is fortunate, but not necessarily new. If you can demonstrate small flight biases for instrument pairs that are consistent over a campaign, now THAT would represent "major progress".

→ While there is always room for further progress, we do believe that the agreement between ChiWIS and FLASH found in this campaign is very promising. The new Table 3 (thank you) is helpful to show these results quantitatively in a snapshot. Flight-to-flight variations do exist, but are small, and mean biases are < 5% for all flights (clear and cloudy). But so as not to oversell our results we have removed the term "major progress" from this sentence.